# Long-range skin Josephson supercurrent across a van der Waals ferromagnet

Guojing Hu[1,5], Changlong Wang[1,5], Shasha Wang[1], Ying Zhang[1], Yan Feng[1], Zhi Wang [2] ✉, Qian Niu[3], Zhenyu Zhang [4] & Bin Xiang [1] ✉

The emerging field of superconducting spintronics promises new quantum device architectures without energy dissipation. When entering a ferromagnet, a supercurrent commonly behaves as a spin singlet that decays rapidly; in contrast, a spin-triplet supercurrent can transport over much longer distances, and is therefore more desirable, but so far has been observed much less frequently. Here, by using the van der Waals ferromagnet $Fe_3GeTe_2$ (F) and spin-singlet superconductor $NbSe_2$ (S), we construct lateral Josephson junctions of **S/F/S** with accurate interface control to realize long-range skin supercurrent. The observed supercurrent across the ferromagnet can extend over 300 nm, and exhibits distinct quantum interference patterns in an external magnetic field. Strikingly, the supercurrent displays pronounced skin characteristics, with its density peaked at the surfaces or edges of the ferromagnet. Our central findings shed new light on the convergence of superconductivity and spintronics based on two-dimensional materials.

An exciting prospect stemming from the combination of superconductivity and spintronics is the appearance of a spin-polarized supercurrent, capable of transporting not only charge but also a net spin component without any dissipation. However, superconductivity and ferromagnetism are two antagonistic macroscopic orderings[1,2], in which the superconductivity can be destroyed by a strong exchange effect unless the Cooper pairing is of the triplet nature[3–6]. To this end, a robust synergy between superconductivity and spintronics is enabled by inducing spin-triplet pair correlations at superconductor/ferromagnet interfaces[7–12]. One compelling experimental evidence of spin-triplet supercurrents was reported in a half-metallic ferromagnet of $CrO_2$ between two spin-singlet superconductors of NbTiN[13]. Subsequently, a series of experiments reported more evidence of spin-triplet pairing in Josephson junctions with inhomogeneous metallic ferromagnets[14–16] or metallic magnets[17–19] as the barrier materials.

To date, the earlier efforts have been primarily focused on materials of bulk nature. More recently, the rising stars of layered or van der Waals (vdW) materials, such as the two-dimensional (2D) ferromagnets $CrI_3$[20,21], $Cr_2Ge_2Te_6$[22,23], $Fe_3GeTe_2$ (F)[24,25], and superconductor $NbSe_2$ (S)[26,27], provide unprecedented opportunities to explore spin-triplet supercurrents. In particular, by engineering vdW S/F interfaces, 2D superconducting spintronics can be developed without the constraint of lattice matching[28–30].

In this paper, we report the first construction of a lateral Josephson junction composed of a vdW metallic ferromagnet ($Fe_3GeTe_2$) laterally connected between two layered spin-singlet superconductors ($NbSe_2$), and use the S/F/S architecture to sustain skin Josephson supercurrents with long-range nature. The observed supercurrent across the ferromagnet can extend over 300 nm, and exhibits double-slit quantum interference patterns in an external magnetic field. Strikingly, the supercurrent displays pronounced skin characteristics, as signified by its peaked densities at the surfaces or edges of the ferromagnet. The present study provides a new platform for facile generation of spin-triplet supercurrents in superconducting spintronics based on 2D materials.

[1]Department of Materials Science & Engineering, CAS Key Lab of Materials for Energy Conversion, Anhui Laboratory of Advanced Photon Science and Technology, University of Science and Technology of China, 230026 Hefei, China. [2]School of Physics, Sun Yat-sen University, 510275 Guangzhou, China. [3]School of Physical Sciences, University of Science and Technology of China, 230026 Hefei, China. [4]International Center for Quantum Design of Functional Materials (ICQD), University of Science and Technology of China, 230026 Hefei, China. [5]These authors contributed equally: Guojing Hu, Changlong Wang. ✉e-mail: wangzh356@mail.sysu.edu.cn; binxiang@ustc.edu.cn

## Results

### S/F/S lateral Josephson junction

$Fe_3GeTe_2$ is a van der Waals ferromagnetic metal with a Curie temperature of 200 K as measured in our experiment, and it has a strong perpendicular magnetic anisotropy with an easy magnetization direction along the c-axis. Its hexagonal crystal structure with a space group of $P6_3/mmc$ is composed of a layered $Fe_3Ge$ substructure that is sandwiched between two layers of Te atoms (Fig. 1b). The $Fe_3Ge$ substructure is comprised of two types of Fe atoms: Fe I and Fe II. Fe I sites are fully occupied while Fe II sites are only partially occupied. A frustrated triangular structure is created by the Fe I atoms, resulting in spins of the Fe atoms forming a noncoplanar structure (Fig. 1b). This noncoplanar structure acts as a fictitious magnetic field due to the presence of a nontrivial Berry phase in real space[31–34]. $NbSe_2$ is known to be a van der Waals superconductor with spin-singlet pairing correlation ($T_c = 7.0$ K).

Figure 1c shows an optical image of a lateral vdW Josephson junction of S/F/S prepared by a dry-transfer method (details presented in the Methods section). The width w and length $L_j$ (edge-to-edge separation between $NbSe_2$) of the junction channel are ~2 μm and 300 nm, respectively. The thicknesses of the $NbSe_2$ and $Fe_3GeTe_2$ are 25 and 22 nm, respectively. The whole device is shielded from air contamination by a top layer of hBN. To study the electrical properties of the S/F/S, we first measure the temperature dependence of the four-probe resistance ($R$–$T$) at zero magnetic field ($R = V/I_{app}$, $I_{app} = 10$ μA). With decreasing temperature, the $R$–$T$ result (Fig. 1d) shows an abrupt resistance drop towards zero with three transition temperatures, with the first two corresponding to the superconducting transition of $NbSe_2$ (6.9 K) and the proximity-induced superconducting transition of the S/F bilayers at each end of the S/F/S (5.4 K). When the temperature is further lowered, the observed tail of the S/F/S resistance drop is well fitted by the Berezinskii–Kosterlitz–Thouless (BKT) model with a BKT temperature of 3.7 K, strongly indicating that the zero-resistance state is approached as vortex-antivortex pairs bind together[35,36].

### Long-range supercurrent with spin-triplet nature

The measured current–voltage curve of the S/F/S (Fig. 2a) shows a clear superconducting feature of a sizable Josephson critical current $I_c$ as large as 100 μA at zero voltage, which nevertheless is much smaller than the critical current of pristine $NbSe_2$ (Supplementary Fig. 1). The S/F/S's $I$–$V$ curve does not exhibit a strong hysteretic behavior, which indicates that the junction is in an overdamped regime with a low resistance-capacitance product[17,37]. When the applied current exceeds $I_c$, the $I$–$V$ curve exhibits a linear behavior crossing the origin and displays a normal state resistance $R_n$ of 16 Ω.

To further study the Josephson supercurrent, we measure the critical current as a function of temperature in the S/F/S with different channel lengths $L_j$. The $T$-dependent critical current (Fig. 2b) can be fitted by $I_c(T) \approx I_c(0)(1 - \frac{T}{T_c})^{\alpha}$, where $I_c(0)$ is the critical current at zero-temperature, and $T_c$ is the S/F/S superconducting transition temperature[37]. We can derive the zero-temperature critical currents $I_c(0)$ to be 126.03, 90.37, 64.73 μA for $L_j = 260, 280, 300$ nm by applying different values of $\alpha$, respectively. Again, these critical currents are one or two orders of magnitude smaller than those in pristine $NbSe_2$ (Supplementary Fig. 1). As the channel length $L_j$ increases, the zero-temperature critical current $I_c(0)$ shows a tendency to decay. This is attributed to the reduction of proximity-induced Cooper pairs in the longer $Fe_3GeTe_2$ barrier. When the channel length increased to 450 nm, the $I$–$V$ curve of the device exhibits a linear behavior, indicating that the superconducting current disappears completely (Fig. 2c). $L_j$-dependent characteristic voltage $I_cR_n$ product of the junctions is shown in the inset of Fig. 2c. The fitting of an exponential decay function $\exp(-\frac{L_j}{\xi})$ shows an estimated coherence length of $\xi = 227 \pm 18$ nm. However, by using $\xi_{singlet}^{FM} \approx \sqrt{\frac{\hbar D}{2E_{ex}}}$[8,9,38,39], where $D$ and $E_{ex}$ are, respectively, the $Fe_3GeTe_2$ electron diffusion coefficient and exchange energy, we obtain the coherence length of the spin-singlet supercurrent in the $Fe_3GeTe_2$ layer to be ~3.5 nm, much shorter than the $L_j$. Therefore, the long-range nature of the supercurrents across the ferromagnetic $Fe_3GeTe_2$ layer observed in the S/F/S might be related to

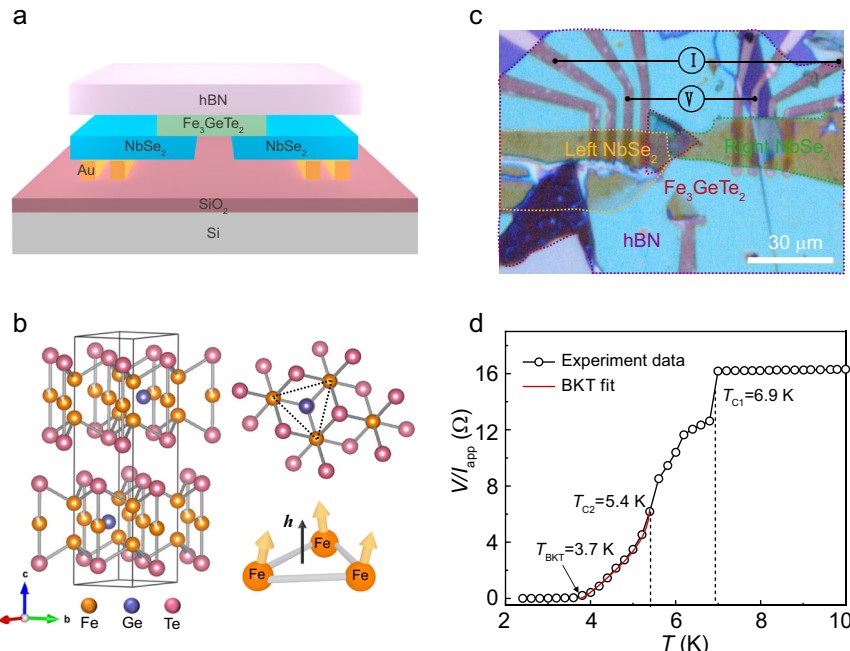

**Fig. 1 | S/F/S device architecture and temperature-dependent resistivity.**
**a** Schematic illustration of a S/F/S lateral Josephson junction. **b** Schematic diagram of the atomic structure of $Fe_3GeTe_2$, where showing noncoplanar spin textures with a fictitious field of $h$ in a frustrated triangular lattices. **c** Optical image of the S/F/S device. **d** Temperature dependence of the S/F/S resistance by a four-terminal measurement and the applied current ($I_{app}$) is 10 μA. Three

transitions are identified, with the first two $T_{c1} = 6.9$ K from superconducting $NbSe_2$, and $T_{c2}$-5.4 K from the proximity-induced superconducting transition of the S/F bilayers at each end of the S/F/S. The red solid line represents the BKT transition using the Halperin–Nelson equation for fitting, which gives the third transition with a BKT temperature $T_{BKT} = 3.7$ K. Source data are provided as a Source Data file.

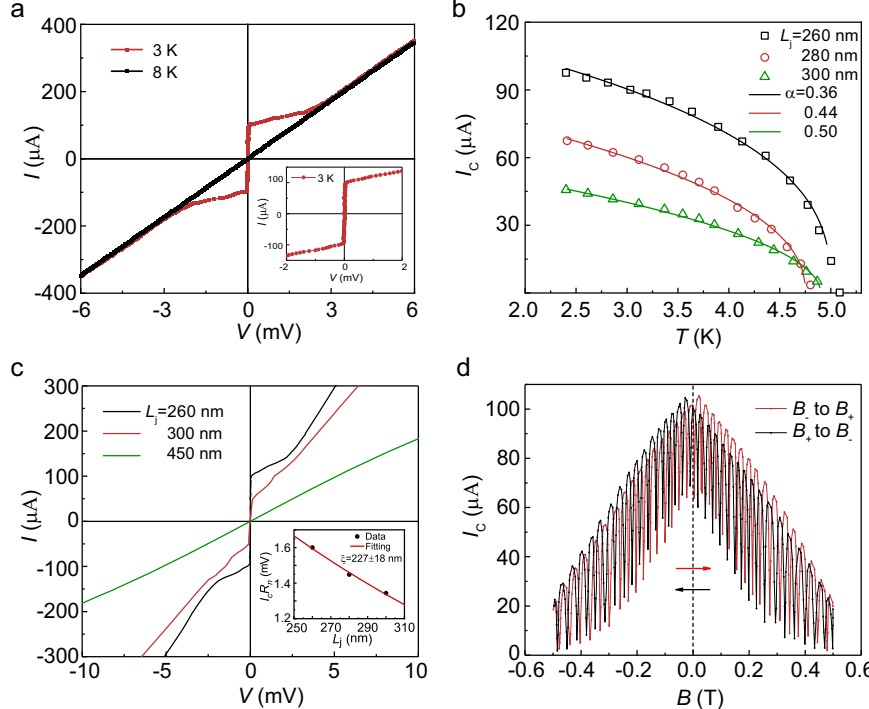

**Fig. 2 | Transport characteristics of the S/F/S devices. a** Current–voltage (*I*–*V*) curves for S/F/S Josephson junction at temperatures of 3 and 8 K under zero magnetic field, showing the $I_c$-100 µA at 3 K, and the inset shows the magnified plot of *I*–*V* curve around zero voltage. **b** Critical current $I_c$ as a function of temperature with different $L_j$ of 260, 280, 300 nm. The solid lines are theoretical fitting curves. **c** Current–voltage (*I*–*V*) curves for Josephson junctions with different channel lengths at temperatures of 3 K under zero magnetic field. $L_j$-dependent characteristic voltage $I_cR_n$ product of the junctions in the inset. **d** Critical current $I_c$ as a function of the external *in-plane* magnetic field with forward and reverse directions of field sweep. The arrows represent the direction of the scanning magnetic field. Source data are provided as a Source Data file.

the proximity-induced spin-triplet superconducting correlations. To further prove that the long-range Josephson supercurrents is directly related to the nature of Fe₃GeTe₂, we replace the Fe₃GeTe₂ layer with a ferromagnet metal of Fe₀.₂₅TaS₂ as a barrier to contract NbSe₂/Fe₀.₂₅TaS₂/NbSe₂ lateral junction with the channel length of 200 nm. There is no Josephson supercurrent observed in such NbSe₂/Fe₀.₂₅TaS₂/NbSe₂ junction (Supplementary Fig. 2).

**Skin characteristics in spin-triplet supercurrent**

To further investigate the quantum interference of S/F/S supercurrent, we measure the response of $I_c$ to an external *in-plane* magnetic field that is perpendicular to the supercurrent channel. Figure 2d shows a double-slit quantum interference behavior in the amplitude of $I_c$ as the magnetic field is varied, which is different from the phenomenon in a self-stacking NbSe₂ junction (Supplementary Fig. 3). When the magnetic field is along the *z*-axis (Fig. 3a), the magnetic-field dependence of the critical current $I_c(B)$ exhibits a pure sinusoidal double-slit pattern (Fig. 3b), indicating that the supercurrent density along the *y*-axis is characterized by only two conductive edge-channels, which exhibit significant depletion of bulk carriers. The full widths at half maxima of these two edge-channels are -1.80 nm and 1.33 nm, as determined by Gaussian fitting (Fig. 3c). When the magnetic field is along the *y*-axis direction (Fig. 3d), the $I_c(B)$ exhibits a mixed single-slit Fraunhofer and double-slit interference pattern (Fig. 3e). The supercurrent density along the *z*-axis (extracted from Fig. 3e) is relatively uniform, but with two pronounced peaks (Fig. 3f), indicating surface-dominated transport along the two edges of the Fe₃GeTe₂ strip.

In addition, the quantum oscillation of $I_c(B)$ at both the *out-of-plane* and *in-plane* magnetic fields shows a clear hysteresis feature with an offset at the first maximum of the supercurrent from zero magnetic field, caused by the spontaneous magnetic flux from Fe₃GeTe₂ finite magnetization[13,40]. Typically, the total enclosed magnetic flux within a

Josephson junction is the result of both the external flux generated by the magnetic field and an intrinsic flux induced by the magnetization of the barrier[13,40,41], which means that the offset is dependent on the barrier moment. Therefore, when we measure the $I_c(B)$ with forward and reverse directions of field scanning, the Fe₃GeTe₂ magnetization loop induces this offset from zero magnetic field. The $I_c(B)$ at the *in-plane* magnetic field has a larger offset than at the *out-of-plane* magnetic field.

## Discussion

Before closing, we conjecture on the possible mechanism for the observed long-range spin-triplet supercurrent. The notable aspect of the present work is that this long-range Josephson supercurrent exhibits a striking skin feature that is distinct from conventional bulk channels. We propose two possible physical mechanisms in the present system that might be responsible for this skin feature. First, we argue that both top and bottom surface of the Fe₃GeTe₂ layer has a mirror symmetry breaking, which can induce a Rashba spin-orbit coupling. In principle, the interplay of this Rashba spin-orbit coupling, the ferromagnetism of the bulk Fe₃GeTe₂, and the s-wave superconductivity of NbSe₂ may induce a formation of 2D topological superconductivity on the surface of the Fe₃GeTe₂. In this case, the top and bottom surfaces of the Fe₃GeTe₂ could support an effective chiral *p*-wave topological superconductivity that was first proposed by Fu and Kane[42]. Of course this topological state is only a theoretical speculation that is consistent with interference patterns and symmetry restrictions. The experimental identification of topological superconductivity in the present system is certainly an exciting direction for further studies.

The second possible mechanism involves the noncoplanar structure (Fig. 1b) of the Fe atoms, which was predicted to produce a fictitious magnetic field. When a conventional s-wave superconductor is in

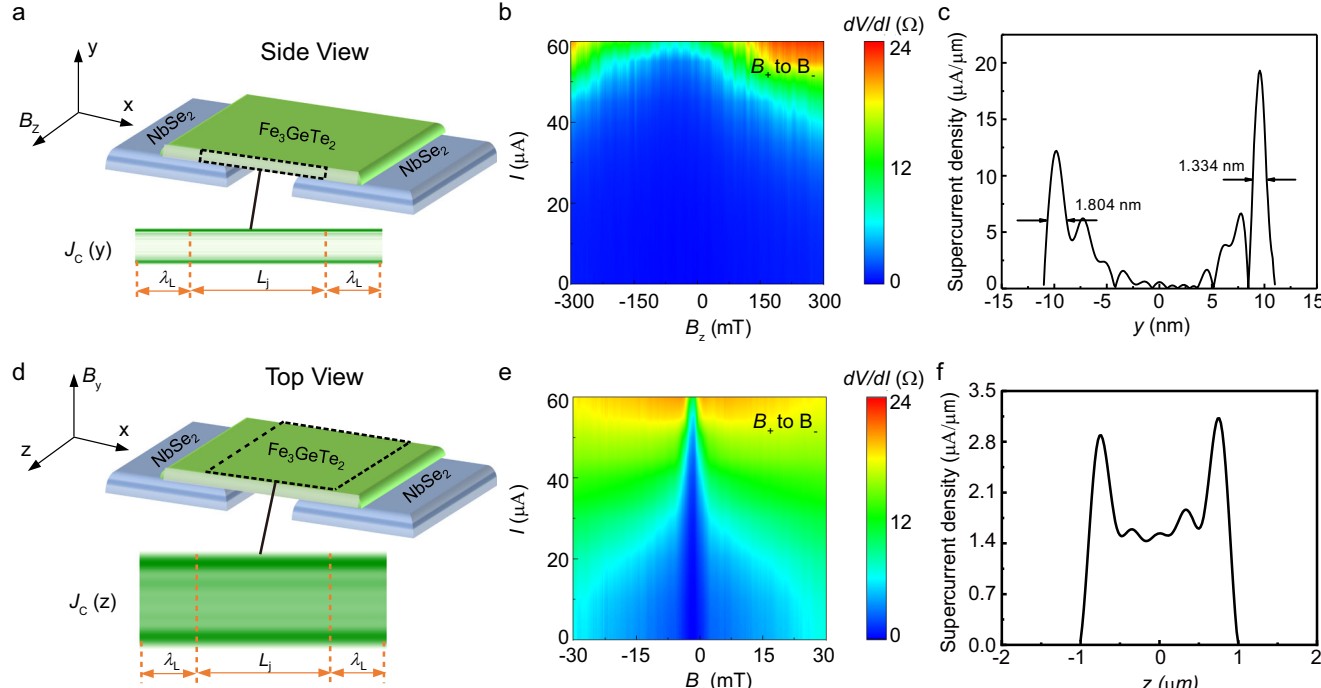

**Fig. 3 | Skin effects in the supercurrent density of S/F/S Josephson junctions.**
**a** Schematic of the S/F/S with the magnetic field along the $z$-axis direction.
**b** Differential resistance map across the junction at 3 K, showing a double-slit interference pattern. **c** Distribution of supercurrent density along the $y$-axis obtained from the inverse Fourier transform of the data in **b**. The $Fe_3GeTe_2$ layer thickness is 22 nm. **d** Schematic of the S/F/S junction with the magnetic field along the $y$-axis direction. **e** Differential resistance map across the junction at 3 K, showing a mixed single-slit and double-slit interference pattern. **f** Distribution of supercurrent density along the $z$-axis obtained from the inverse Fourier transform of the data in **e**, showing that the supercurrent can flow relatively more uniformly across the junction along the $x$-axis, but still peaked at the surfaces (now the two edges). The channel width $w$ is 2 μm. Source data are provided as a Source Data file.

contact with magnetic inhomogeneity, inhomogeneous magnetizations facilitates the conversion of spin-singlet Cooper pairs into spin-triplet pairs via spin-mixing and spin-rotation processes[9,11,43]. In the microscopic processes of entering the interface region to $Fe_3GeTe_2$, the electrons of singlet Cooper pairs of $NbSe_2$ experience spin-dependent interface phase shifts[44,45], then only a mixture of spin-singlet and spin-triplet ($S_z = 0$) Cooper pairs can be present[46] (named spin-mixing). Thus, the noncoplanar spin textures of $Fe_3GeTe_2$ within frustrated triangular lattices[31] have the ability to orient the spin-quantization axis to align with the local magnetization direction. Then, this can flip one of the spins in the triplet component ($S_z = 0$)[45]. As a result, the spin-triplet pairs ($S_z = \pm1$), consisting of the same spin pairs with equal amplitude and opposite sign, are produced with a long decay length[46], which is insensitive to the pair-breaking effect from the Zeeman field, but limited by a thermal coherence length[23]. However, noncoplanar spins formed in a system of triangular iron atoms have only been proposed in theory[47], while no direct experimental evidence has yet been reported about the observation of this noncoplanar structure in the bulk $Fe_3GeTe_2$. Then experimental verification of the noncoplanar structure in current $Fe_3GeTe_2$ system also requires extensive further study. In conclusion, whether the long-range superconducting current has a spin-triplet state and the physical mechanism of the skin behavior of the superconducting current require more in-depth theory and experiments in the future.

In summary, we have experimentally demonstrated that a long-range supercurrent is generated at van der Waals interfaces in S/F/S lateral Josephson junctions. The decay length of the proximity-induced Cooper pairs with the same spin across the ferromagnet $Fe_3GeTe_2$ can be as long as 300 nm. More strikingly, the supercurrent density distribution profile provides the first evidence of skin feature of long-range spin-triplet superconductivity. Our work provides a new platform with superior control of interface properties for future

exploration of novel physical properties and potential device applications of 2D superconducting spintronics.

## Methods

### Single-crystal growth

The $Fe_3GeTe_2$ single crystals were grown by a chemical vapor transport (CVT) method. The details of the growth are specified in our previous work[48]. The $NbSe_2$ single crystals were also obtained by the CVT method. The synthesis process for the single crystal is shown in Supplementary Fig. 4. High-purity Niobium and Selenium (molar ratio Nb:Se = 1:2) in stoichiometric amounts were mixed with iodine (4 mg/$cm^2$) as a transport agent before being sealed into an evacuated quartz tube. We raise the temperatures of the source and growth zones in the two-zone tubular furnace to 850 °C and 750 °C, respectively, at a rate of 5 °C/min and maintained at 850/750 °C for 10 days. Then the furnace was left to cool naturally to room temperature after the growth process, and the resulting shiny crystals were collected.

### Characterization techniques

The crystal structures of $Fe_3GeTe_2$ and $NbSe_2$ were characterized by X-ray diffraction. The magnetic properties of the single crystals were measured by a SQUID (MPMS3), and the thickness of the exfoliated samples was measured by an atomic force microscope.

### Electrode processing

Electrodes with different geometries were patterned on a 300-nm-thick $SiO_2$/Si substrate by ultraviolet lithography with Ti/Au (8/25 nm) metal deposited on the surface of substrate through magnetron sputtering. After the lift-off process, electrodes with different geometries were obtained for heterostructure transfer process. In this work, parallel electrodes with a spacing of 30 μm were used.

## Heterostructure assembly by a dry-transfer technique

Mechanical exfoliation and dry-transfer processes were carried out within a glove box that maintained highly controlled levels of $O_2$ and $H_2O$ (< 0.01 ppm) in order to minimize their presence. The fabrication process is shown in Supplementary Fig. 5. Firstly, we transferred the desired $NbSe_2$ nanoflake onto the left side of the parallel electrodes. Then we stacked a layer of $NbSe_2$ onto the right side of the parallel electrodes without touching the left $NbSe_2$. A layer of $Fe_3GeTe_2$ was placed on the gap position bridging the two layers of $NbSe_2$. Finally, we used a layer of hBN to encapsulate heterostructure for protection (Supplementary Fig. 6).

## Electric transport measurements

The electric transport measurements were performed using a variable temperature cryostat provided by Oxford Instruments, which allowed for accurate temperature control from 300 K down to 2.4 K, and the magnetic field strength ranges from −1.4 T to 1.4 T. The resistance vs. temperature ($R–T$) curves and differential conductance resistance ($dV/dI$) were measured using a Keithley 6221 and 2182 A as the current source and voltmeter, respectively. Longitudinal electrical resistance and Hall resistance were acquired using a standard five-point measurement technique with the current set to be 10 μA.

## Data processing

Due to the asymmetry in our nanoflake samples, the measured Hall resistance was mixed with the longitudinal magnetoresistance. Typically, We processed the Hall resistance data $R_{xy}$ by using $(R_{xy}(+B) − R_{xy}(−B))/2$ to remove the contribution from the longitudinal magnetoresistance, and utilized $(R_{xx}(+B) + R_{xx}(−B))/2$ to eliminate the contribution from the Hall magnetoresistance to obtain longitudinal magnetoresistance $R_{xx}$. Besides, we define the temperature at which the resistance reaches half of the normal resistance as the transition temperature ($T_c$). We differentiate the $I–V$ curve to obtain the $I–dV/dI$ curve. The position at half of the difference between the maximum value and the minimum value in the $dV/dI$ curve is defined as the Josephson critical current (Supplementary Fig. 7).

## Data availability

The source data underlying all figures are available as a Source Data file provided with this paper. All relevant data are available from the corresponding author upon request. Source data are provided with this paper.

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

## Acknowledgements

This work was supported by Innovation Program for Quantum Science and Technology (2021ZD0302800), National Synchrotron Radiation Laboratory (KY2060000177), and National Natural Science Foundation of China (12174453, 12234017, 92165204). This research was partially carried out at the USTC Center for Micro and Nanoscale Research and Fabrication.

## Author contributions

G.H. carried out all the experiments with assistance from C.W., S.W., Y.Z., Y.F., Z.Z., Q.N., B.X., W.Z., G.H. analyzed the data. Z.Z., Q.N., B.X., and G.H. prepared and edited the manuscript. This project was initiated and supervised by B.X., Q.N., and Z.Z. All the authors have read the manuscript and agreed with its content.

## Competing interests

The authors declare no competing interests.
