## [Peer Review File · Nature Communications]

Long-range skin Josephson supercurrent across a van der Waals ferromagnetREVIEWER COMMENTS

Reviewer #1 (Remarks to the Author):

In their manuscript entitled 'Long-range skin Josephson supercurrent across a van der Waals ferromagnet', Hu et al., reported the findings of long-ranged Josephson supercurrents through a lateral junction geometry based on a vdW ferromagnet Fe₃GeTe₂ with noncoplanar magnetic structure, which is related to the spin-triplet correlations. Also, they found that the supercurrents display some skin characteristics with their densities peaked at the surfaces or edges of the barrier. The fabrication of such a narrow channel length junction via stacking is not easy to accomplish, and the observation of considerable supercurrents is clear and solid when channel length exceeds the pair-breaking length. However, I wondered if their results of variations in channel lengths could give a quantification of decay length of triplet-pairing states, and I was also confused about the origin of its skin characteristics. Below are the issues that I would appreciate the authors considering.

1. The authors give 3 channel lengths results (260, 280, 300 nm) in Fig. 2c, while the orders are very close and no fittings here represent the tendency of exponential decay of its coherence length. As the 300nm-junction has a sufficient magnitude of supercurrent, what would we expect in longer channel devices? At what distance will the supercurrent drop to almost negligible?
2. The skin features they showed are mainly from the double-slit quantum interference behavior, but I didn't see any interpretations on why it happens in this 'special' vdW ferromagnet. Is it related to the 2D lamellar nature of such a nanoflake or the geometry of this junction, as their barrier is put on top of those two superconductors? Considering that the F-layer is indeed sandwiched into two SC-layers vertically, will the skin features still be observed?
3. The hysteresis of $I_c(B)$ they showed in Fig. 2d is abnormal since the two $I_c(B)$ curves from opposite field-sweeping directions don't coincide at larger fields. I think the authors should provide the rest data until the barrier's magnetization is approaching saturation.
4. The difference between the extracted L_{eff} and the edge-to-edge channel length L_j might be accounted for by the wrong estimation of the period of oscillation, which is extracted from the interference pattern mixed with supercurrent distributions both from the bulk and edge channels. I think it's unreasonable to do such an analysis.

In addition, there are some unclear statements needed to be improved:

1. Para 3, line 2, 'sandwiched' is confusing, as the Fe₃GeTe₂ is placed on the top of two NbSe₂.
2. Para 4, line 1, the Curie temperature of bulk Fe₃GeTe₂ (~200K) is lower than the reported values (~220K), so I suggest the authors could refer here to their own measurements in supplementary materials.
3. Para 6, line 4, 'The S/F/S does not exhibit a strong hysteretic behavior...' should be corrected to 'The S/F/S's IV does not exhibit a strong hysteretic behavior...'
4. Para 8, line 4, '...ensuring that the two NbSe₂ layers are not short circuited' may not be true. To my knowledge, the NbSe₂ self-stacked Josephson junction could also show the apparent oscillation behavior of $I_c(B)$.
5. Para 10, line 8, the value of magnetic hysteresis of $I_c(B)$ at the out-of-plane and the in-plane field should be given here to compare with its perpendicular magnetic anisotropy.
6. Fig 3b and 3e, the field sweep direction should be indicated, and the $I_c(B)$ patterns along the opposite direction should be presented in supplementary materials.
7. Supplementary Fig. 13, I find it unnecessary to show no supercurrent results of NbSe₂/Fe_{0.25}TaS₂/NbSe₂ junction, as the authors did not emphasize whether there are noncollinear or noncoplanar magnetic structures in Fe_{0.25}TaS₂ either in the main text or supplementary materials.

Reviewer #2 (Remarks to the Author):

Remarks to the Author:

The authors have observed a long-range skin supercurrent with spin-polarised triplet nature in NbSe₂/Fe₃GeTe₂/NbSe₂ van der Waals (vdW) lateral Josephson junctions. The observed supercurrent across the Fe₃GeTe₂ barrier can extend over 300 nm, which is approximately 2 orders of magnitude larger than expected for a spin-unpolarised singlet supercurrent. They have also found SQUID-like magnetic-field interference patterns, which are presumably due to stronger proximity coupling through the surfaces or edges of the Fe₃GeTe₂ barrier.

While I find the results quite interesting, there are several issues that must be addressed before further consideration of this manuscript in Nature Communications.

1) In the section of S/F/S lateral Josephson junction (on page 4), the authors describe that “a frustrated triangular structure is formed by the Fe I atoms, so the spins of the Fe atoms form a noncoplanar structure (Fig. 1b) that acts as a fictitious magnetic field due to the presence of a nontrivial Berry phase...”

In this regard, I am curious how large the misalignment angle between neighbouring spins of the Fe atoms is and how strong the corresponding fictitious magnetic field (in k-space) is. I think the authors should quantify the misalignment angle and fictitious field strength as these are directly connected to the singlet-to-triplet pair conversion efficiency.

2) If I have a close look at I-V curves of JJs around zero voltage (Fig. 2a, Fig. S11), there may exist a non-zero voltage component even at 3 K.

For clarity, magnified plots need to be presented in each inset. In fact, I have noticed from the T-dependent junction's resistance (Fig. 1d) that the spontaneous nucleation of vortex and antivortex pairs within a Josephson barrier possibly gives rise to a non-vanishing resistance even below the junction's T_c. The authors should also describe how the Josephson critical current is defined in Method.

3) From the characteristic voltage I_cR_n, corresponding to Josephson coupling energy, versus barrier length L_j data and using an exponential decay function, one can estimate the decay length of Josephson supercurrents (or coherence length) through a Fe₃GeTe₂ barrier.

I would suggest the authors replace the I_c(L_j) plot in Fig. 2c with a I_cR_n(L_j) plot and try to fit it with the exponential fit. In addition, how spin-flip scattering and spin-orbit scattering in the Fe₃GeTe₂ barrier influence the overall coherence length and how these scatterings can be excluded from the dominant relaxation/dephasing mechanisms of spin-polarised triplet pair correlation need to be briefly discussed in Main text.

4) On page 6, a hysteresis feature with an offset at the zero-order maximum of the supercurrent from zero magnetic field is attributed to the spontaneous magnetic flux from Fe₃GeTe₂ finite magnetization.

I agree on this point but the offset of ~75 mT in Fig. 3b does not seem to match a coercive field (~400 mT) of the perpendicularly magnetized Fe₃GeTe₂ in Fig. S9d. The authors should discuss possible origins of this discrepancy.

5) Regarding the interpretation of SQUID-like magnetic-field interference patterns (Fig. 3), I wonder how the spontaneous and/or field-assisted nucleation of vortex and anti-vortex pairs modifies the spatial transverse uniformity of Josephson supercurrent density. If a Josephson barrier is packed by a certain number of the vortex and antivortex pairs, the transverse supercurrent density distribution can, in principle, be inhomogeneous and localized.

To confirm that the SQUID-like interference patterns (Fig. 3) result from topological properties of Fe₃GeTe₂, I think the author should provide direct evidence [e.g. using angle-resolved photoemission spectroscopy (ARPES), see a relevant experiment arXiv:2112.11285].

6) In Fig. 3f, the FFT-estimated supercurrent density is plotted in the z-axis up to ± 1 μm. Yet, I am not sure why the estimated FGT width (2 μm) seems much narrower than the actual dimension (10-15 μm wide) of the fabricated device presented in Fig. 1c.

7) It would be great if the authors could investigate whether or not the Josephson supercurrent in a NbSe₂/Fe₃GeTe₂/NbSe₂ vertical geometry is also long-ranged. I think this experiment is quite important as it enables one to identify which mechanism (fictitious field versus spin-orbit coupling) is responsible for the surprisingly long-ranged supercurrent (~300 nm at 3 K) through a topological vdW ferromagnet Fe₃GeTe₂.

8) I am also curious how magnetic skyrmions and chiral spin-texture in the Fe₃GeTe₂ (Nano Lett. 2020, 20, 868–873, Adv. Mater. 2022, 34, 2108637) play a role in the creation of long-range proximity coupling. Do the author think the skyrmions and chiral spin-texture have a minor contribution?

I hope it helps improve the overall quality of the paper to be better suited for the high impact journal of Nature Communications.

Reviewer #3 (Remarks to the Author):

This manuscript reports experimental measurements of supercurrent through a Josephson junction containing a piece of a van der Waals ferromagnet, Fe₃GeTe₂. From the dependence of the junction critical current on magnetic field, the authors deduce that the supercurrent density is concentrated in the very top and bottom layers of the ferromagnet. Furthermore, the authors claim that the supercurrent must be carried by spin-triplet electron pairs, due to its long-range nature (up to 300 nm).

The experimental is novel, and the data are interesting. The in-depth Supplement discusses many important aspects of the sample characterization and data analysis. But I have one strong criticism of the manuscript, plus several additional comments and criticisms.

1) While the evidence for the supercurrent and its unusual spatial distribution are solid, the argument for the spin-triplet nature of the supercurrent is much weaker. There is an alternative explanation, which is much simpler, namely that the top and bottom layers of the exfoliated VdW material are not magnetic. Even in the case of very strong magnetic materials, there is often a magnetic “dead” layer at the surface, especially if it is in contact with another material. It would not be surprising if Fe₃GeTe₂ has similar nonmagnetic dead layers at its surfaces.

The authors have been careful not to mention “spin-triplet” in the title of their manuscript. It would be easy to fix the abstract: don’t mention spin-triplet until the very end, as a speculation. The body of the manuscript would have to be substantially changed to downplay the interpretation in terms of spin-triplet supercurrent.

2) The manuscript contains 53 references, but unfortunately several of the most important ones are missing. The second sentence of the abstract reads: “When entering a ferromagnet, a supercurrent commonly behaves as a spin singlet that decays rapidly; in contrast, a spin-triplet supercurrent can transport over much longer distances, and is therefore more desirable, but so far has been observed much less frequently [1-3].” References [1] and [3] are highly appropriate here, but reference [2] is a theory paper. More importantly, the authors have failed to cite two of the three groups who have contributed the most to the experimental literature in this field, namely the groups at Michigan State University and the University of Leiden. (The third key group at Cambridge authored reference [3].) At the very least, the authors should cite Khaire et al., Phys. Rev. Lett. 104, 137002 (2010) and Anwar et al., Phys. Rev. B 82, 100501 (2010). There are plenty of other spin-triplet papers from those two groups, as well as other groups around the world.

Here are a few more places where the references are inadequate:

i) At the end of the first paragraph on page 3, we read, “Subsequently, a series of experiments reported more evidence of spin triplet pairing in Josephson junctions with inhomogeneous metallic ferromagnets [2,9,14] or chiral metallic magnets [15-17] as the barrier materials.” Reference [9] is again a theory paper, while reference [14] reports tunneling experiments on an S/F structure with no

discussion of spin-triplet pairs. (Reference [14] is a seminal paper, but it is unrelated to the topic being discussed here.) Regarding references [16] and [17], I do not remember any discussion of chirality in those papers, but perhaps I am mistaken.

ii) On line 5 of the first paragraph in the section, "Long-range supercurrent with spin-triplet nature", the authors cite references [15,35]. I believe they meant to cite [36] rather than [35].

iii) The second sentence of the discussion cites references [9,11,47,48]. The first three of those are appropriate. The last one is an interesting paper, but I do not remember any discussion of spin-triplet pairs in that paper.

3) In the description of the sample, the authors write, "In this paper, we report the first construction of a lateral Josephson junction composed of a vdW metallic ferromagnet (Fe_3GeTe_2) sandwiched between two layered spin-singlet superconductors (NbSe_2)..." That is an incorrect description. If the Fe_3GeTe_2 were sandwiched between two NbSe_2 layers, then the supercurrent would flow vertically. A lateral junction and a sandwich junction are two opposite geometries – one sample cannot be both.

4) The discussion of the sample resistance vs temperature makes no sense without specifying the measurement current. (It is mentioned only in the Methods section as being 10 microAmp.) The current-voltage relation for a Josephson junction is highly nonlinear, so characterizing it as a resistance is misleading. Similarly, modeling the temperature dependence of that resistance with the BTK theory seems odd to me; I was under the impression that the BTK theory was designed for bulk 2D superconductors rather than for Josephson junctions. A more straightforward interpretation of so-called "BTK" temperature of 3.7 K shown in Figure 1(d) is that that is the temperature where the critical current I_c becomes larger than the measurement current. If a different measurement current were chosen, then the zero-resistance temperature would also change accordingly.

4) It is very difficult to ascertain the critical current from Figures 3(b) and (e). Only when I saw Supplement Figure 10 was I able to see what is happening in 3(b).

5) The last sentence before the Discussion reads, "The $I_c(B)$ at the in-plane magnetic field has a larger offset than at the out-of-plane magnetic field, consistent with perpendicular magnetic anisotropy of the Fe_3GeTe_2 layer [45,46]." That argument sounds backwards to me. Perpendicular magnetic anisotropy should cause a larger magnetic remanence, and larger hysteresis, when the field is oriented out-of-plane.

6) Supplementary Note 2 ends with an estimate of the spin-triplet decay length of 40 nm. And yet the data shown in Figure 2(c) surely have a decay length much longer than that. On the other hand, the three junctions studied surely have different cross-sectional areas (i.e. the lateral dimensions of the NbSe_2 electrodes are different), so plotting their critical current on the same plot is meaningless.

7) The previous point brings up another issue: how accurately do the authors know the spacing between the NbSe_2 electrodes in their junctions? The best picture we are shown is in the inset to Supplemental Figure 12(a). Given the resolution of that figure, it is unclear how well one can determine the electrode spacing.

8) When the authors plot the spatial distribution of the supercurrent, they claim that the widths of the two sharp peaks at the edges of the sample are meaningful, being determined by the data via the Fourier transform process. But those widths may have a contribution from the finite field range of the data. They may also have a contribution from the fact that the electrode spacing is not constant, due to the unevenly shaped NbSe_2 electrodes.

9) In Supplemental Note 2, point (2), one reads that the Hall resistivity is linear in magnetic field H . But the data shown in Supplemental Figure 9(d) are linear only at temperatures above the Curie temperature. At what temperature are the authors carrying out this calculation?

10) In Supplemental Figure 1(a), the low temperature region should be expanded. There is no need to show the high-temperature data where nothing happens. Figure 1(b) needs arrows to show the sweep direction in the hysteresis region. Also, how is B_{c1} defined?

11) In Supplemental Figure 6, there appears to be a missing peak around $2\theta = 45$ degrees. Are the authors sure that their indexing scheme is correct? Note the contrast with Supplemental Figure 8(b), where the peaks are nearly uniformly spaced in angle.

Responses to reviewers' comments

(MS# NCOMMS-22-25728-T)

We thank all three reviewers for investing their valuable time and effort on reviewing our manuscript and we appreciate the comments that they provided which helped us to clarify, strengthen, and improve our manuscript. We believe that our modifications have fully addressed the referees concerns and that our revised manuscript is currently organized in such a way to reflect more clearly on the power and impact of our results. Based on the responses below, we have also updated two figures in the revised manuscript, four figures and one table in the Supplementary Information, to address the comments. Our point-by-point responses to the referees are as follows.

Reviewer #1

1. Comment: The authors give 3 channel lengths results (260, 280, 300 nm) in Fig. 2c, while the orders are very close and no fittings here represent the tendency of exponential decay of its coherence length. As the 300 nm-junction has a sufficient magnitude of supercurrent, what would we expect in longer channel devices? At what distance will the supercurrent drop to almost negligible?

Response: *In Fig. 2c, the three channel length results are from three different devices with different channel length L_j , width, and thickness. The three devices have different cross-section areas. As a result, it may not be an appropriate way to draw the L_j -dependent superconducting critical current I_c . It is challenging to fabricate three identical devices with all the same dimension parameters except for different channel lengths, by this mechanical exfoliation and dry transfer method. Therefore, we present a table as shown in Table R1 with all listed dimension parameters of the three devices instead of the L_j -dependent I_c plot, which roughly shows a decay tendency with increasing channel length L_j .*

We fabricated the devices with a longer channel length as shown in Figure R1. The I - V curve of the device with a channel length of ~ 450 nm exhibits a linear behavior, and the superconducting current disappears completely. We have revised the following table and figure in our revised manuscript and Supporting Information.

Table R1 Parameters of different S/F/S junctions: I_c is the superconducting critical current; L_j is the channel length; w is the width of the cross-sectional areas; t is the thickness of the Fe_3GeTe_2 (FGT).

Device	I_c (μA)	L_j (nm)	w (μm)	t (nm)
1	100	260	0.55	14
2	68	280	0.47	12
3	45	300	2	22
4	0	450	0.29	11

Figure R1 Current-voltage (I - V) curves for Josephson junctions with different channel lengths at temperatures of 3 K under zero magnetic field.

2. Comment: The skin features they showed are mainly from the double-slit quantum interference behavior, but I didn't see any interpretations on why it happens in this 'special' vdW ferromagnet. Is it related to the 2D lamellar nature of such a nanoflake or the geometry of this junction, as their barrier is put on top of those two superconductors? Considering that the F-layer is indeed sandwiched into two SC-layers

vertically, will the skin features still be observed?

Response: *We thank the referee for this comment concerning the physical mechanism behind the skin features for the Josephson current distribution. From a theoretical point of view, two possible mechanisms in the present system could be responsible for this skin feature.*

It is obvious that both the top and bottom surface of the FGT layer have a mirror symmetry break, which can induce a Rashba spin-orbit coupling. The first mechanism would be a formation of 2D topological superconductivity on the surface of the FGT due to the interplay of the ferromagnetism of the bulk FGT, the s-wave superconductivity of NbSe₂, and the Rashba spin-orbit coupling on the top and bottom surfaces of FGT from the inversion symmetry breaking. In this mechanism, the top and bottom surfaces of the FGT support an effective topological superconductivity that was first proposed by Fu and Kane¹. In our own opinion, this surface topological superconducting state is quite a reasonable possibility. If this mechanism is verified with further experiments, we would have an interesting new platform for Majorana zero modes. However, the experimental verification of topological superconductivity and Majorana modes is a well-known difficult task and is certainly beyond the scope of current work. We are likely to peruse along this direction in our further works.

The second possible mechanism involves the noncoplanar structure (Fig. 1b) of the Fe atoms, which was predicted to be producing a fictitious magnetic field from a nontrivial Berry phase in the energy band of the FGT. In principle, this Berry phase in the band structure would induce a nontrivial spin-texture in the Fermi surface of the FGT. Then the spin-singlet Cooper pair may be converted to spin-triplet pairing in the FGT and induce a surface supercurrent.

We note that these two mechanisms are both unique to the surface of the FGT. Rashba spin-orbit coupling is missing in the bulk of the FGT layer because FGT has a centrosymmetric structure and there is no inversion symmetry breaking in the bulk. In addition, noncoplanar spins formed in a triangular lattice from iron atoms have only been proposed in theory², while no direct experimental evidence has been reported about observations of this noncoplanar structure in bulk. Therefore, we argue that the

FGT favors the supercurrent flowing with the skin features, resulting from the surface spin-orbit coupling. Our device is a “lateral” Josephson junction. The wording of “a vdW metallic ferromagnet (Fe_3GeTe_2) sandwiched between two layered spin-singlet superconductors ($NbSe_2$)” was misleading. We apologize for the confusion and have fixed it in the revised manuscript.

3. Comment: The hysteresis of $I_c(B)$ they showed in Fig. 2d is abnormal since the two $I_c(B)$ curves from opposite field-sweeping directions don't coincide at larger fields. I think the authors should provide the rest data until the barrier's magnetization is approaching saturation.

Response: *The hysteresis of $I_c(B)$ shown in Figure 2d is due to the finite coercivity of FGT interlayer. When we extend to larger fields, the two $I_c(B)$ curves from opposite field-sweeping directions shown in Figure R2 exhibit a concordance in larger fields. We may think that the hysteresis should disappear or the two $I_c(B)$ curves from opposite field-sweeping directions should coincide when the external magnetic field approaches the FGT saturation state. However, in our data (Figure R2), the hysteresis disappears and the two $I_c(B)$ curves coincide in a lower magnetic field. We believe that the superconductivity indeed can effectively weaken but not eliminate the ferromagnetism of the FGT layer through a proximity effect because of the interaction between the superconductivity and ferromagnetism that has been widely reported³. As a result, the saturated magnetization in the superconducting FGT layer is smaller than that in a pristine FGT layer. Figure R2c shows that the magnetization of the FGT layer with an $NbSe_2$ layer underneath is smaller than that of the FGT layer without an $NbSe_2$ layer underneath characterized by magnetic force microscopy (MFM), experimentally confirming that the superconductivity can weaken the magnetization of the FGT layer through the proximity effect. Therefore, we observed a concordance between the two $I_c(B)$ curves from opposite field-sweeping directions in a lower external field.*

Figure R2 (a) Critical current I_c as a function of the external in-plane magnetic field with forward and reverse directions of field sweep. The arrows represent the direction of the scanning magnetic field. (b) Schematic illustration of FGT/NbSe₂ heterostructure characterized by MFM. (c) Magnetization distribution of the FGT layer across a boundary region of the heterostructure with and without the NbSe₂ underneath, obtained by MFM in 0 T magnetic field.

4. Comment: The difference between the extracted L_{eff} and the edge-to-edge channel length L_j might be accounted for by the wrong estimation of the period of oscillation, which is extracted from the interference pattern mixed with supercurrent distributions both from the bulk and edge channels. I think it's unreasonable to do such an analysis.

Response: We agree with the reviewer's comment. Figure 3e shows that the interference pattern is a mixture of Fraunhofer and SQUID-like modes caused by supercurrent distributions both from the bulk and edge channels. Thus, it could be less accurate to calculate the effective channel length by estimating the period of oscillation in the mixed modes. In addition, although Fig. 2d and 3b exhibit a pure sinusoidal double-slit pattern, the overlaid regions between FGT and NbSe₂ at micrometer-scale also have certain contributions to the oscillation pattern. Therefore, it is hard to accurately calculate the effective channel length from this SQUID-like oscillation pattern. Further studies are needed in the future.

5. Comment: Para 3, line 2, ‘sandwiched’ is confusing, as the Fe_3GeTe_2 is placed on the top of two NbSe_2 .

Response: *The use of “sandwiched” was misleading. We have revised the sentences “In this paper, we report the first construction of a lateral Josephson junction composed of a vdW metallic ferromagnet (Fe_3GeTe_2) sandwiched between two layered spin-singlet superconductors (NbSe_2)...” into “In this paper, we report the first construction of a lateral Josephson junction composed of a vdW metallic ferromagnet (Fe_3GeTe_2) laterally connected between two layered spin-singlet superconductors (NbSe_2)...”. We have also carefully checked the English writing and modified the descriptions in our revised manuscript.*

6. Comment: Para 4, line 1, the Curie temperature of bulk Fe_3GeTe_2 (~200K) is lower than the reported values (~220 K), so I suggest the authors could refer here to their own measurements in supplementary materials.

Response: *In the revised manuscript, we have updated the sentence to “ Fe_3GeTe_2 is a van der Waals ferromagnetic metal with a Curie temperature of 200 K as measured in our experiment.”*

7. Comment: Para 6, line 4, ‘The S/F/S does not exhibit a strong hysteretic behavior...’ should be corrected to ‘The S/F/S’s IV does not exhibit a strong hysteretic behavior...’.

Response: *We have revised this as you recommend. We have also carefully checked the related English text and updated the information in our revised manuscript.*

8. Comment: Para 8, line 4, ‘...ensuring that the two NbSe_2 layers are not short circuited’ may not be true. To my knowledge, the NbSe_2 self-stacked Josephson junction could also show the apparent oscillation behavior of $I_c(\text{B})$.

Response: *It has been reported in the literature that self-stacking of NbSe_2 can also form a Josephson junction due to different order parameters of the exfoliated NbSe_2^4 . The I-V curve of the self-stacking NbSe_2 Josephson junction (Figure R3a) shows a clear*

hysteresis behavior, and the response of I_c to an in-plane external magnetic field in Figure R3b shows a Fraunhofer pattern. The Josephson behaviors in the self-stacking NbSe₂ Josephson junction differ from the observed phenomenon in our S/F/S heterostructure. We have corrected the sentence to be “Fig. 2d shows a double-slit quantum interference behavior in the amplitude of I_c as the magnetic field is varied, which is different from the phenomenon in the self-stacking NbSe₂ junction (Supplementary Fig. 2).” We have modified the descriptions in our revised manuscript.

Figure R4 The Josephson behaviors in the self-stacking NbSe₂ Josephson junction. (a) Current–voltage (I – V) curve measured by sweeping the current at 2 K. Arrows indicate the sweep direction of the current. (b) Contour plot of V as a function of I and $B_{//}$ at 2 K, showing a Fraunhofer pattern⁴.

9. Comment: Para 10, line 8, the value of magnetic hysteresis of $I_c(B)$ at the out-of-plane and the in-plane field should be given here to compare with its perpendicular magnetic anisotropy.

Response: The $I_c(B)$ at the in-plane magnetic field has an offset of 63.8 mT, which is much larger than that of the out-of-plane magnetic field (1.7 mT), consistent with perpendicular magnetic anisotropy of the Fe₃GeTe₂ layer. We have added the related information in our revised manuscript.

10. Comment: Fig 3b and 3e, the field sweep direction should be indicated, and the

$I_c(B)$ patterns along the opposite direction should be presented in supplementary materials.

Response: *The field sweep directions in Fig 3b and 3e have been indicated, and the $I_c(B)$ patterns along the opposite direction have been presented in supplementary materials. We have updated Figure R4 in our revised manuscript.*

Figure R4 *a, Differential resistance map across the junction at 3 K with the magnetic field scanning from negative to positive, showing a sinusoidal double-slit interference pattern. b, Differential resistance map across the junction at 3 K with the magnetic field scanning from negative to positive, showing a single-slit interference pattern.*

11. Comment: Supplementary Fig. 13, I find it unnecessary to show no supercurrent results of NbSe₂/Fe_{0.25}TaS₂/NbSe₂ junction, as the authors did not emphasize whether there are noncollinear or noncoplanar magnetic structures in Fe_{0.25}TaS₂ either in the main text or supplementary materials.

Response: *Fe_{0.25}TaS₂, a 2H-TaS₂ based magnetic ion Fe intercalated transition metal dichalcogenide⁵, exhibits a ferromagnetic behavior with the Curie temperature T_c about 150 K. From the atomic structure model shown in Figure R5, we can see that the structure of Fe_{0.25}TaS₂ is very different from that of Fe₃GeTe₂, so we carried out the study of NbSe₂/Fe_{0.25}TaS₂/NbSe₂ junction to illustrate that ferromagnetic metal without topological property cannot generate a long-range superconducting current. We have added the related information in Supporting Materials.*

Figure R5 *a*, Schematic diagram of the atomic structure of $\text{Fe}_{0.25}\text{TaS}_2$. *b*, Schematic illustration of $\text{NbSe}_2/\text{Fe}_{0.25}\text{TaS}_2/\text{NbSe}_2$ lateral Josephson junction. *c*, Current-voltage (I - V) curves for Josephson junction at temperatures of 3 and 8 K under zero magnetic fields.

Reviewer #2

1. Comment: In the section of S/F/S lateral Josephson junction (on page 4), the authors describe that “a frustrated triangular structure is formed by the Fe I atoms, so the spins of the Fe atoms form a noncoplanar structure (Fig. 1b) that acts as a fictitious magnetic field due to the presence of a nontrivial Berry phase....”

In this regard, I am curious how large the misalignment angle between neighbouring spins of the Fe atoms is and how strong the corresponding fictitious magnetic field (in k -space) is. I think the authors should quantify the misalignment angle and fictitious field strength as these are directly connected to the singlet-to-triple pair conversion efficiency.

Response: *From a theoretical point of view, the fictitious field should be proportional*

to the vector product of the three Fe spins $\mathbf{S}_1 \cdot (\mathbf{S}_2 \times \mathbf{S}_3)$. However, its amplitude in the momentum space is determined by the detailed band structure of the FGT and requires a very detailed theoretical analysis that is beyond the scope of current work. We cannot quantify the misalignment angle and fictitious field strength at this stage for the following reasons:

(1) The Fe_3Ge substructure of the FGT structure contains two types of Fe atoms: Fe I and Fe II. The Fe I sites are fully occupied, and the Fe II sites are partially occupied. The two Fe sites cause a complicated exchange interaction, which requires very complicated calculations.

(2) The Curie temperature of FGT depends on the FGT layer thickness^{6,7}, which indicates that the interlayer magnetic interactions also play an important role in determining the size of the fictitious field. Therefore, it is hard to quantify the interlayer magnetic interactions.

(3) We did an extensive literature search but found no reports of quantitative studies on the FGT misalignment angle and fictitious field strength so far.

Actually, another likely mechanism for the long-range supercurrent may be even more appealing, resulting from Rashba spin-orbit coupling effect on the FGT surface. Because of the interplay of the ferromagnetism of the bulk FGT, the *s*-wave superconductivity of NbSe_2 , and the Rashba spin-orbit coupling, 2D topological superconductivity can be formed on the FGT surface¹, which supports a skin effect feature.

2. Comment: If I have a close look at *I-V* curves of JJs around zero voltage, there may exist a non-zero voltage component even at 3 K.

For clarity, magnified plots need to be presented in each inset. In fact, I have noticed from the T-dependent junction's resistance (Fig. 1d) that the spontaneous nucleation of vortex and antivortex pairs within a Josephson barrier possibly gives rise to a non-vanishing resistance even below the junction's T_c . The authors should also describe how the Josephson critical current is defined in Method.

Response: *The magnified plots of I-V curves at zero voltage (Fig. 2a and Fig. S11)*

have been presented in insets (Figure R6a and b). The magnified inset in Figure R7a shows no non-zero voltage components, while the magnified inset of Figure R6b shows a small non-zero voltage component. We define the temperature at which the resistance reaches half of the normal resistance as the transition temperature (T_c), so the T -dependent junction's resistance (Fig. 1d) has a non-vanishing resistance below the junction's T_c . The junction reaches a zero-resistance state until the temperature decreases to T_{BKT} as shown in Figure 1d. Further, we differentiate the I - V curve to obtain the I - dV/dI curve as shown in Figure R6c. The position at half of the difference between the maximum value and the minimum value in the dV/dI curve is defined as the Josephson critical current. We discuss this method to define the Josephson critical current in our revised manuscript.

Figure R6 a, Current-voltage (I - V) curves for $S/F/S$ Josephson junction with channel length of 260 nm at temperatures of 3 and 8 K under zero magnetic field; the inset shows the magnified plots of I - V curves around zero voltage. b, Current-voltage (I - V) curves for $S/F/S$ Josephson junction with channel length of 300 nm at temperatures of 3 and 8 K under zero magnetic field; the inset shows the magnified plots of I - V curves around zero voltage. c, The I - dV/dI curve of Josephson junction at temperature 3 K under zero magnetic field; the position at half of the difference between the maximum value and the minimum value in the I - dV/dI curve is defined as the Josephson critical current I_c .

3. Comment: From the characteristic voltage $I_c R_n$, corresponding to Josephson coupling energy, versus barrier length L_j data and using an exponential decay function, one can estimate the decay length of Josephson supercurrents (or coherence length)

though a Fe_3GeTe_2 barrier.

I would suggest the authors replace the $I_c(L_j)$ plot in Fig. 2c with a $I_c R_n(L_j)$ plot and try to fit it with the exponential fit. In addition, how spin-flip scattering and spin-orbit scattering in the Fe_3GeTe_2 barrier influence the overall coherence length and how these scatterings can be excluded from the dominant relaxation/dephasing mechanisms of spin-polarized triplet pair correlation need to be briefly discussed in Main text.

Response: *In Fig. 2c, the three channel length results are from three different devices with different channel length L_j , width, and thickness. The three devices have different cross-section areas. As a result, it may not be an appropriate way to draw the L_j -dependent superconducting critical current I_c . It is challenging to fabricate three identical devices with all the same dimension parameters except for different channel lengths, by this mechanical exfoliation and dry transfer method. Therefore, we present a table as shown in Table R2 with all listed dimension parameters of the three devices instead of the L_j -dependent I_c plot, which roughly shows a decay tendency with increasing channel length L_j .*

At this time, our experimental results (Figure R1) show no supercurrent is observed in the lateral junction of $\text{NbSe}_2/\text{Fe}_3\text{GeTe}_2/\text{NbSe}_2$ when its channel length is increased to 450 nm, while there is an obvious long-range supercurrent in the junction when the channel length is below 300 nm. Here, we believe that a topological superconductivity on the FGT layer surface dominates the lateral Josephson junction transport, resulting from the spin-orbit coupling. Unfortunately, we do not know how spin-flip scattering and spin-orbit scattering in the Fe_3GeTe_2 barrier influence the overall coherence length and how these scatterings can be excluded from the dominant relaxation/dephasing mechanisms of spin-polarized triplet pair correlation. Further studies are needed in the future. For example, in the same junction device, a different channel length is required to provide a chance to study the decay length of Josephson supercurrents by an exponential decay function fitting of its I-V curves. This device could be fabricated by e-beam lithography technique to obtain s-wave superconducting-Nb electrode patterns with different distance between neighboring Nb electrodes, after which a layer of FGT could be transferred onto the Nb electrodes,

which can maintain all the device parameters except the channel length (different Nb-electrode distances).

The e-beam lithography technique can fabricate electrode patterns with less than 100 nm distance between adjacent electrode. Once the decay length is obtained, it can be compared with the theoretically calculated coherence length of the spin-triplet, which could probably distinguish the topological supercurrent from the pure spin-triplet supercurrent. Unfortunately, we do not have access to e-beam lithography. With our current device fabrication processes, it is difficult to achieve identical devices with different channel lengths or to fabricate one device with different channel lengths. More experimental research is needed.

Table R2 Parameters of different S/F/S junctions: I_c is the superconducting critical current; L_j is the channel length; w is the width of the cross-section areas; t is the thickness of the Fe_3GeTe_2 .

Device	I_c (μA)	L_j (nm)	w (μm)	t (nm)
1	100	260	0.55	14
2	68	280	0.47	12
3	45	300	2	22
4	0	450	0.29	11

4. Comment: On page 6, a hysteresis feature with an offset at the zero-order maximum of the supercurrent from zero magnetic field is attributed to the spontaneous magnetic flux from Fe_3GeTe_2 finite magnetization.

I agree on this point but the offset of ~ 75 mT in Fig. 3b does not seem to match a coercive field (~ 400 mT) of the perpendicularly magnetized Fe_3GeTe_2 in Fig. S9d. The authors should discuss possible origins of this discrepancy.

Response: We think that this discrepancy originates from the interaction between the superconductivity and ferromagnetism in Fe_3GeTe_2 . The induced superconducting state in Fe_3GeTe_2 can weaken the ferromagnetism of Fe_3GeTe_2 , which results in a smaller

coercive field in superconducting Fe_3GeTe_2 than in pristine Fe_3GeTe_2 . We construct a bilayer heterostructure of Fe_3GeTe_2 on the top of $NbSe_2$ (Figure R7a) and map out the real-space magnetization distribution of the $Fe_3GeTe_2/NbSe_2$ heterostructure at 4 K by utilizing magnetic force microscopy (MFM). As shown in Figure R7b, the MFM image taken of the Fe_3GeTe_2 with an $NbSe_2$ layer underneath at zero magnetic field shows much weaker ferromagnetism (much less contrast) than the Fe_3GeTe_2 layer without an $NbSe_2$ layer underneath, indicating that the superconductivity indeed effectively weakens but does not eliminate the ferromagnetism of the Fe_3GeTe_2 layer. Therefore, we observe a smaller offset in our lateral Josephson junction compared to the coercive field of the pristine Fe_3GeTe_2 . We have added this discussion to our revised manuscript.

Figure R7 (a) Schematic illustrate of FGT/ $NbSe_2$ heterostructure characterized by MFM. (b) Magnetization distribution of the FGT layer across a boundary region of the heterostructure with and without the underlying $NbSe_2$ by MFM at 0 T magnetic field.

5. Comment: Regarding the interpretation of SQUID-like magnetic-field interference patterns (Fig. 3), I wonder how the spontaneous and/or field-assisted nucleation of vortex and anti-vortex pairs modifies the spatial transverse uniformity of Josephson supercurrent density. If a Josephson barrier is packed by a certain number of the vortex and antivortex pairs, the transverse supercurrent density distribution can, in principle, be inhomogenous and localized.

To confirm that the SQUID-like interference patterns (Fig. 3) result from topological

properties of Fe_3GeTe_2 , I think the author should provide direct evidence [e.g. using angle-resolved photoemission spectroscopy (ARPES), see a relevant experiment arXiv:2112.11285].

Response: *Indeed, the spontaneous or field-induced vortices may modify the supercurrent distribution and change the Fraunhofer pattern. This phenomenon of a Josephson vortex has been a heated research area recently and we would be excited to find signals indicating the existence of a Josephson vortex in our devices. Unfortunately, there is no such signature in our experimental results shown in Fig. 3b. From the device feature (Fig. 3a), the thickness of the FGT layer is ~ 20 nm. The calculation (Supplementary Note S2) shows that the coherence length is ~ 40 nm. Therefore, we believe that no vortex or anti-vortex could form because the core size of the vortex is comparable to the coherence length, which is larger than the channel thickness of the FGT layer. As a result, in the SQUID-like interference pattern when the magnetic field is along the z-direction (Fig. 3a), there could be no nucleation of vortex and anti-vortex pairs.*

In addition, Stolyarov et al. have reported a strong asymmetry and distortions observed in the magnetic-field interference pattern in a Josephson junction device because of the generation of the vortex^{8,9}. However, our observed SQUID-like interference pattern is quite symmetrical with respect to the maximum point, which is qualitatively different from the irregular interference patterns expected and observed for junctions with Josephson vortices. With these considerations, we feel that the Josephson vortex is not the dominant mechanism behind the skin supercurrent observed in the SQUID-like interference pattern. However, from the symmetry of Fig. 3b and 3e, we cannot completely exclude the existence of vortex-antivortex pairs, because of the observation of the hysteresis shift at zero magnetic field in these interference patterns. In particular, in Fig. 3e, when the magnetic field is along the y direction, some vortex generation could occur because the core size of such a vortex could survive with that channel length and channel width.

The electronic structure of Fe_3GeTe_2 has been widely studied using angle-resolved photoemission spectroscopy (ARPES)^{10,11}. Jun Sung Kim et al. have reported that a

robust orbital-driven nodal line structure in Fe_3GeTe_2 (Fig. R8) is confirmed by the theoretical calculations and ARPES characterizations, resulting from a spin-orbital coupling, and the topological nodal lines are protected by crystalline symmetries. In addition, they found that the band crossing along the nodal lines can produce a large Berry curvature. Therefore, the SQUID-like interference patterns could be attributed to the topological properties of Fe_3GeTe_2 .

Figure R8 *a*, ARPES intensity at the E_F in the k_x - k_y plane—the red hexagon indicates the Brillouin zone, which exhibits electron Fermi surfaces at the K point and hole Fermi surfaces at the Γ point. The ARPES spectra were taken on a single crystal of Fe_3GeTe_2 . *b*, The calculated Fermi surface, based on density functional theory bands, is plotted for the Γ plane, and the black lines indicate the directions of the ARPES cuts presented in *c*–*e*. *c*, ARPES spectra along the K – K cut is well reproduced in the k_z -integrated dynamical mean-field theory (DMFT) spectral function. Note that a relatively large k_z dispersion suppresses the expected spectra from the majority-spin bands near the K point, consistent with experiments. *d,e*, ARPES spectra and k_z -integrated DMFT

spectral function along $K-\Gamma-K$ (d) and $K-M-K$ (e) cuts match with each other¹⁰.

6. Comment: In Fig. 3f, the FFT-estimated supercurrent density is plotted in the z-axis up to $\pm 1 \mu\text{m}$. Yet, I am not sure why the estimated FGT width ($2 \mu\text{m}$) seems much narrower than the actual dimension ($10\text{-}15 \mu\text{m}$ wide) of the fabricated device presented in Fig. 1c.

Response: *From the optical image shown in Figure 1c, the irregular shapes of the exfoliated FGT and NbSe₂ lead to irregular contact edges. Although the width of the adjacent metal-electrodes is about $6 \mu\text{m}$, we defined the width of the nearest segment as the width of the channel, which is about $2 \mu\text{m}$ as it has the strongest electric field. Therefore, the FFT-estimated supercurrent density is plotted in the z-axis up to $\pm 1 \mu\text{m}$.*

7. Comment: It would be great if the authors could investigate whether or not the Josephson supercurrent in a NbSe₂/Fe₃GeTe₂/NbSe₂ vertical geometry is also long-ranged. I think this experiment is quite important as it enables one to identify which mechanism (fictitious field versus spin-orbit coupling) is responsible for the surprisingly long-ranged supercurrent ($\sim 300 \text{ nm}$ at 3 K) through a topological vdW ferromagnet Fe₃GeTe₂.

Response: *We fabricated the NbSe₂/Fe₃GeTe₂/NbSe₂ vertical heterojunctions with different thicknesses of Fe₃GeTe₂; the schematic structure and optical photos are shown in Figure R9a and b. Figure R9c shows the I-V curves of Josephson junctions with different FGT thicknesses measured at 3 K . When the thickness of FGT is 5 nm , the junction shows an obvious Josephson critical current of $\sim 200 \mu\text{A}$. When the thickness of the FGT increases to 8 nm , a large non-zero voltage component appears in the I-V curve, which indicates that the Josephson supercurrent in a NbSe₂/Fe₃GeTe₂/NbSe₂ vertical structure does not have long-range behavior. Therefore, from the above results, we may conclude that in our lateral Josephson junction, probably no long-range supercurrent flows in the FGT bulk layer, and the surface topological supercurrent dominates the junction transports because of the surface spin-orbit coupling effect.*

Figure R9 a, Schematic illustration of the NbSe₂/FGT/NbSe₂ vertical Josephson junction. b, Optical image of the vertical Josephson junction. c, Current-voltage (I-V) curves for vertical Josephson junctions with different FGT thickness at temperatures of 3 K under zero magnetic field.

8. Comment: I am also curious how magnetic skyrmions and chiral spin-texture in the Fe₃GeTe₂ (Nano Lett. 2020, 20, 868–873, Adv. Mater. 2022, 34, 2108637) play a role in the creation of long-range proximity coupling. Do the author think the skyrmions and chiral spin-texture have a minor contribution?

Response: From a theoretical point of view, we propose two possible mechanisms in the present system that could be responsible for this long-range supercurrent.

It is obvious that both the top and bottom surface of the FGT layer have a mirror symmetry breaking, which can induce a Rashba spin-orbit coupling. Accordingly, the first mechanism would be a formation of 2D topological superconductivity on the FGT surface, due to the interplay of the ferromagnetism of the bulk FGT, the s-wave superconductivity of NbSe₂, and the Rashba spin-orbit coupling on both top and bottom surfaces of the FGT from the inversion symmetry breaking. In this mechanism, the top and bottom surfaces of the FGT support the effective topological superconductivity that was proposed by Sau et al¹.

The second possible mechanism involves the noncoplanar structure (Fig. 1b) of the Fe atoms, which was predicted to produce a fictitious magnetic field from a nontrivial Berry phase in the energy band of the FGT. In principle, this Berry phase in the band structure would induce a nontrivial spin-texture in the Fermi surface of the FGT. As proved by the literature¹², Fe₃GeTe₂ is a strong uniaxial magnet with an easy axis along

the $c(z)$ direction, and the Fe atoms form a rather frustrated triangular structure as shown in Fig. 1b. Hence during the in-plane magnetization process, the spins of Fe atoms could form a noncoplanar structure that acts as a fictitious magnetic field \mathbf{h} .

In both mechanisms, the common factor is a spin texture, either a spin-texture in momentum space due to SOC, or a spin-texture in real space due to noncoplanar structure. Therefore, we believe that the reported magnetic skyrmions and chiral spin-texture in the Fe_3GeTe_2 could play an important role in the creation of long-range proximity coupling. The FGT noncoplanar spin textures or skyrmion precession may act as a spin-rotation source to align the spin-quantization axis with the local magnetization direction, thus, flipping one of the spins in the triplet component ($S_z=0$). As a result, the spin-triplet pairs ($S_z=\pm 1$) with equal amplitude and opposite signs are generated with a long decay length.

Reviewer #3

1. Comment: While the evidence for the supercurrent and its unusual spatial distribution are solid, the argument for the spin-triplet nature of the supercurrent is much weaker. There is an alternative explanation, which is much simpler, namely that the top and bottom layers of the exfoliated vdW material are not magnetic. Even in the case of very strong magnetic materials, there is often a magnetic “dead” layer at the surface, especially if it is in contact with another material. It would not be surprising if Fe_3GeTe_2 has similar nonmagnetic dead layers at its surfaces.

Response: *Magnetic dead layers generally appear in conventional magnetic oxide films. As the film thickness decreases, the ferromagnetism and metallicity of the film decrease rapidly. When the film thickness is less than a certain critical thickness, the film exhibits insulating properties, and the ferromagnetism is greatly reduced, which is called the “dead layer effect”. However, when the FGT material is exfoliated to a monolayer, it has the strongest stable ferromagnetism⁶, which exhibits a layer-thickness dependence of ferromagnetism with behavior opposite to the tendency observed in conventional films. This is a special property for van der Waals ferromagnetic materials^{13,14}. Another important issue is that Fig. 3b shows a long-range skin*

supercurrent at the top and bottom surfaces of the FGT layer. The thickness of the FGT layer is ~ 20 nm. If there is a magnetic “dead” layer on the FGT surface, the spin-singlet supercurrent has trouble reaching and flowing on the top surface of the FGT layer. Therefore, we believe no magnetic “dead” layer exists on the FGT layer surfaces.

2. Comment: The authors have been careful not to mention “spin-triplet” in the title of their manuscript. It would be easy to fix the abstract: don’t mention spin-triplet until the very end, as a speculation. The body of the manuscript would have to be substantially changed to downplay the interpretation in terms of spin-triplet supercurrent.

Response: *The experimental results show direct evidence of long-range skin supercurrent, but we have no direct evidence to prove the “spin-triplet” behavior in our observed supercurrent. Therefore, we have fixed the abstract to avoid using “spin-triplet” until the end as speculation. We reworded similarly in the body of the manuscript.*

3. Comment: The manuscript contains 53 references, but unfortunately several of the most important ones are missing. The second sentence of the abstract reads: “When entering a ferromagnet, a supercurrent commonly behaves as a spin singlet that decays rapidly; in contrast, a spin-triplet supercurrent can transport over much longer distances, and is therefore more desirable, but so far has been observed much less frequently [1-3].” References [1] and [3] are highly appropriate here, but reference [2] is a theory paper. More importantly, the authors have failed to cite two of the three groups who have contributed the most to the experimental literature in this field, namely the groups at Michigan State University and the University of Leiden. (The third key group at Cambridge authored reference [3].) At the very least, the authors should cite Khaire et al., Phys. Rev. Lett. 104, 137002 (2010) and Anwar et al., Phys. Rev. B 82, 100501 (2010). There are plenty of other spin-triplet papers from those two groups, as well as other groups around the world.

Response: *We carefully checked the cited references in the manuscript and updated the*

citations in our revised manuscript.

4. Comment: Here are a few more places where the references are inadequate:

i) At the end of the first paragraph on page 3, we read, “Subsequently, a series of experiments reported more evidence of spin-triplet pairing in Josephson junctions with inhomogeneous metallic ferromagnets [2,9,14] or chiral metallic magnets [15-17] as the barrier materials.” Reference [9] is again a theory paper, while reference [14] reports tunneling experiments on an S/F structure with no discussion of spin-triplet pairs. (Reference [14] is a seminal paper, but it is unrelated to the topic being discussed here.) Regarding references [16] and [17], I do not remember any discussion of chirality in those papers, but perhaps I am mistaken.

Response: *We carefully checked the cited references in the manuscript and corrected the inappropriate references. We modified “chiral metallic magnets [15-17]” into “metallic magnets [15-17]” and updated the related information in our revised manuscript.*

5. Comment:

ii) On line 5 of the first paragraph in the section, “Long-range supercurrent with spin-triplet nature”, the authors cite references [15,35]. I believe they meant to cite [36] rather than [35].

Response: *We carefully checked the cited references in the manuscript and updated the information in our revised manuscript.*

6. Comment:

iii) The second sentence of the discussion cites references [9,11,47,48]. The first three of those are appropriate. The last one is an interesting paper, but I do not remember any discussion of spin-triplet pairs in that paper.

Response: *We carefully checked the cited references in the manuscript and updated the citations in our revised manuscript.*

7. Comment: In the description of the sample, the authors write, “In this paper, we report the first construction of a lateral Josephson junction composed of a vdW metallic ferromagnet (Fe_3GeTe_2) sandwiched between two layered spin-singlet superconductors (NbSe_2)...” That is an incorrect description. If the Fe_3GeTe_2 were sandwiched between two NbSe_2 layers, then the supercurrent would flow vertically. A lateral junction and a sandwich junction are two opposite geometries – one sample cannot be both.

Response: *The use of “sandwiched” was misleading. We have revised the sentences “In this paper, we report the first construction of a lateral Josephson junction composed of a vdW metallic ferromagnet (Fe_3GeTe_2) sandwiched between two layered spin-singlet superconductors (NbSe_2)...” into “In this paper, we report the first construction of a lateral Josephson junction composed of a vdW metallic ferromagnet (Fe_3GeTe_2) laterally connected between two layered spin-singlet superconductors (NbSe_2)...”. We have also carefully checked other aspects of the English writing while updating our manuscript.*

8. Comment: The discussion of the sample resistance vs temperature makes no sense without specifying the measurement current. (It is mentioned only in the Methods section as being 10 micro A mp.) The current-voltage relation for a Josephson junction is highly nonlinear, so characterizing it as a resistance is misleading. Similarly, modeling the temperature dependence of that resistance with the BTK theory seems odd to me; I was under the impression that the BTK theory was designed for bulk 2D superconductors rather than for Josephson junctions. A more straightforward interpretation of so-called “BTK” temperature of 3.7 K shown in Figure 1(d) is that that is the temperature where the critical current I_c becomes larger than the measurement current. If a different measurement current were chosen, then the zero-resistance temperature would also change accordingly.

Response: *We thank the reviewer for the comment. The measurement current (I_{app}) is 10 μA , and this information now appears in the updated figure caption of Figure 1d. Yes, the BKT theory is designed for bulk 2D superconductors rather than for Josephson*

junctions. However, compared to the traditional vertical Josephson junction, our lateral Josephson junction has a special channel structure with a channel length, width, and thickness of 280 nm, 2 μm , and 22 nm, respectively, which shows a two-dimensional (2D) geometry in the FGT channel layer. Moreover, from the spin-triplet coherence length calculation (Supplementary Note S2), we found that the size of the vortex core is several tens of nanometers, which is smaller than the channel length and width. Therefore, the BKT theory could be available to our system. Figure 1d shows a well-fitted behavior by the BKT model with a BKT temperature of 3.7 K. Also, this well-fitted BKT behavior observed in our lateral Josephson junction further confirms a superconducting current existing in the FGT channel layer.

9. Comment: It is very difficult to ascertain the critical current from Figures 3(b) and (e). Only when I saw Supplement Figure 10 was I able to see what is happening in 3(b).

Response: *As shown in Figure R10, the position at half of the difference between the maximum and minimum values in the I - dV/dI curve is defined as the Josephson critical current I_c , which shows a value of 100 μA . Then we plot the critical current as a function of temperature by obtaining the critical current under different magnetic fields. We have added the method to define the Josephson critical current to our revised manuscript.*

Figure R10 *The I - dV/dI curve of the Josephson junction at temperatures of 3 K under zero magnetic field. The position at half of the difference between the maximum and minimum values in the I - dV/dI curve is defined as the Josephson critical current I_c .*

10. Comment: The last sentence before the Discussion reads, “The $I_c(B)$ at the in-plane magnetic field has a larger offset than at the out-of-plane magnetic field, consistent with perpendicular magnetic anisotropy of the Fe_3GeTe_2 layer [45,46].” That argument sounds backwards to me. Perpendicular magnetic anisotropy should cause a larger magnetic remanence, and larger hysteresis, when the field is oriented out-of-plane.

Response: *As shown in Supplementary Fig. 7, the Fe_3GeTe_2 has perpendicular anisotropy, and its easy axis of magnetization is along the c -axis direction (out-of-plane). When the magnetic field in the positive direction scans to the negative magnetic field above the saturation magnetic field, the magnetic moment of the Fe_3GeTe_2 will reverse at the applied negative magnetic field beyond the coercive field of the Fe_3GeTe_2 . When the applied magnetic field is higher than the saturation magnetic field, the magnetic moments will align in the direction of the negative magnetic field. Since the in-plane saturation magnetic field is larger than the out-of-plane one, a larger negative magnetic field is required to flip the magnetic moment of the Fe_3GeTe_2 , which can cause a larger hysteresis phenomenon. Therefore, the $I_c(B)$ in the in-plane magnetic field has a larger offset than in the out-of-plane magnetic field. We have updated the information in our revised manuscript to clarify this point.*

11. Comment: Supplementary Note 2 ends with an estimate of the spin-triplet decay length of 40 nm. And yet the data shown in Figure 2(c) surely have a decay length much longer than that. On the other hand, the three junctions studied surely have different cross-sectional areas (i.e. the lateral dimensions of the $NbSe_2$ electrodes are different), so plotting their critical current on the same plot is meaningless.

Response: *We believe that our long-range skin supercurrent is a topological supercurrent that can travel a longer distance than the estimated length of 40 nm for pure spin-triplet supercurrent. The formation of 2D topological superconductivity on*

the FGT surface is due to the interplay of the ferromagnetism of the bulk FGT, the s-wave superconductivity of NbSe₂, and the Rashba spin-orbit coupling on the top and bottom surfaces of FGT from inversion symmetry breaking.

The mechanically exfoliated FGT and NbSe₂ with irregular shapes lead to different cross-sectional areas in the different devices. As a result, it may not be an appropriate way to draw the L_j -dependent superconducting critical current I_c . It is challenging to fabricate three identical devices with all the same dimension parameters except for different channel lengths, by this mechanical exfoliation and dry transfer method. Therefore, we present a table as shown in Table R3 with all listed dimension parameters of the three devices instead of the L_j -dependent I_c plot, which roughly shows a decay tendency with increasing channel length L_j .

Table R3 *Parameters of different S/F/S junctions: I_c is the superconducting critical current; L_j is the channel length; w is the width of the cross-section areas; t is the thickness of the Fe₃GeTe₂.*

Device	I_c (μ A)	L_j (nm)	w (μ m)	t (nm)
1	100	260	0.55	14
2	68	280	0.47	12
3	45	300	2	22
4	0	450	0.29	11

12. Comment: The previous point brings up another issue: how accurately do the authors know the spacing between the NbSe₂ electrodes in their junctions? The best picture we are shown is in the inset to Supplemental Figure 12(a). Given the resolution of that figure, it is unclear how well one can determine the electrode spacing.

Response: *There should be an error bar in the measurement of the spacing distance between two NbSe₂ layers. However, we minimized the error bar by taking the average of ten measurements. To confirm our measurement accuracy, we have utilized scanning electron microscopy (SEM) to get a higher magnification image of our device structures. The results shown in Figure R11, prove that the spacing distance of 273 nm measured*

using SEM is very close to the value of 267 nm measured using optical microscopy. Therefore, we believe that the spacing-measurement result has reasonable accuracy. We have updated the information in our revised Supporting Information.

Figure R11 SEM image of cross-sectional areas in NbSe₂/FGT/NbSe₂ junctions with an inset of the corresponding optical image.

13. Comment: When the authors plot the spatial distribution of the supercurrent, they claim that the widths of the two sharp peaks at the edges of the sample are meaningful, being determined by the data via the Fourier transform process. But those widths may have a contribution from the finite field range of the data. They may also have a contribution from the fact that the electrode spacing is not constant, due to the unevenly shaped NbSe₂ electrodes.

Response: In our junction shown in Fig. 3b, the magnetic field is applied along the z-axis. The maximum critical current $I_C(B)$ can be viewed as the magnitude of the Fourier transform of supercurrent density $J_s(y)$, so the $J_s(y)$ can be extracted from the inverse Fourier transform of the experimentally measured $I_C(B)$. From the current density distribution in Figure 3c, it can be seen that the supercurrent distribution along the y-axis has two sharp peaks, which correspond to two conductive edge channels in

the sample. We can obtain the width of the edge channel by fitting the half-width of the peak. However, many factors could cause the deviation of width, including the finite field range of the data, error bars in the extraction of superconducting critical current, and measurement of the electrode spacing. Therefore, the channel width of the superconducting current obtained from the Gaussian fitting of the peak width is only an estimation.

14. Comment: In Supplemental Note 2, point (2), one reads that the Hall resistivity is linear in magnetic field H . But the data shown in Supplemental Figure 9(d) are linear only at temperatures above the Curie temperature. At what temperature are the authors carrying out this calculation?

Response: *The Hall resistivity (ρ_{xy}) of ferromagnetic materials is composed of ordinary resistivity ($R_H H$) and anomalous Hall resistivity (ρ_{AHE})¹⁵: $\rho_{xy} = R_H H + \rho_{AHE} = R_H H + R_A M$, where R_H , R_A are the ordinary and anomalous Hall coefficients. The anomalous part (ρ_{AHE}) is related to the magnetization (M) of the ferromagnetic material instead of the external magnetic field (H). Thus, due to the large anomalous Hall effect of FGT at low temperature, the ordinary Hall resistance is not obvious. As shown in Figure R12, when we present the data under a high field (a field near saturation), the linear relationship between the Hall resistance and the magnetic field can be obtained. Then, the ordinary Hall coefficient can be obtained by a linear fitting. Since the I-V measurements were performed at 3 K, we carried out this calculation at that temperature. When the temperature is above the Curie temperature, the FGT becomes paramagnetic. As a result, the anomalous Hall effect disappear, and it shows a completely linear behavior in the Hall curve. We have updated our manuscript to include this information.*

Figure R12 *The Hall resistance of FGT sheet at high field.*

15. Comment: In Supplementary Figure 1(a), the low temperature region should be expanded. There is no need to show the high-temperature data where nothing happens. Figure 1(b) needs arrows to show the sweep direction in the hysteresis region. Also, how is B_{c1} defined?

Response: *We have expanded the low-temperature region in Supplementary Figure 1(a) and added the sweep direction in the hysteresis region in Supplementary Figure 1(b). The second type of superconductivity has two definite critical fields, namely, the lower critical field B_{c1} and the upper critical field B_{c2} . When the external magnetic field is lower than B_{c1} , the magnetic field is completely expelled from the superconductor (the Meissner state), but starting from B_{c1} , the magnetic field partially penetrates into the superconductor. As the magnetic field increases, the penetration also increases ($-M$ decreases). We define B_{c1} as the minimum point in the M - B curve. When the magnetic field reaches B_{c2} , the superconductor is transformed to the normal state. We have updated the figures and the information in our manuscript and Supporting Information.*

Figure R13 *a*, Temperature dependence of the magnetization of single-crystal NbSe₂ under different cooling conditions. *b*, Magnetization of single-crystal NbSe₂ at 3 K as a function of the out-of-plane magnetic field.

16. Comment: In Supplementary Figure 6, there appears to be a missing peak around $2\theta = 45$ degrees. Are the authors sure that their indexing scheme is correct? Note the contrast with Supplementary Figure 8(b), where the peaks are nearly uniformly spaced in angle.

Response: *The X-ray diffraction pattern of FGT single crystal only shows the characteristic diffraction peak of (00L), indicating that the orientation of FGT single crystal is along the c-axis direction. We missed the peak around $2\theta = 45$ degrees and mislabeled the position of the corresponding diffraction peak. Supplementary Fig. 6 is corrected in our Supporting Materials.*

Figure R14 XRD pattern for the as-grown FGT single crystal. The inset is an optical image of the FGT single crystal.

References

1. Fu, L. & Kane, C. L. Superconducting proximity effect and majorana fermions at the surface of a topological insulator. *Phys. Rev. Lett.* **100**, 096407 (2008).
2. Wang, Y. *et al.* Anisotropic anomalous Hall effect in triangular itinerant ferromagnet Fe_3GeTe_2 . *Phys. Rev. B* **96**, 134428 (2017).
3. Buzdin, A. I. Proximity effects in superconductor-ferromagnet heterostructures. *Rev. Mod. Phys.* **77**, 935 (2005).
4. Yabuki, N. *et al.* Supercurrent in van der Waals Josephson junction. *Nat. Commun.* **7**, 1-5 (2016).
5. Wang, J., Quan, Y., Liu, D., Zou, L. Ferromagnetism in Layered Metallic $\text{Fe}_{1/4}\text{TaS}_2$ in the Presence of Conventional and Dirac Carriers, *Chinese Physics Letters* **37**, 017101 (2020).

6. Tan, C. *et al.* Hard magnetic properties in nanoflake van der Waals Fe₃GeTe₂. *Nat. Commun.* **9**, 1-7 (2018).
7. Deng, Y. *et al.* Gate-tunable room-temperature ferromagnetism in two-dimensional Fe₃GeTe₂. *Nature* **563**, 94-99 (2018).
8. Dremov, V. V. *et al.* Local Josephson vortex generation and manipulation with a Magnetic Force Microscope. *Nat. Commun.* **10**, 1-9 (2019).
9. Roditchev, D. *et al.* Direct observation of Josephson vortex cores. *Nat. Phys.* **11**, 332-337 (2015).
10. Kim, K. *et al.* Large anomalous Hall current induced by topological nodal lines in a ferromagnetic van der Waals semimetal. *Nat. Mater.* **17**, 794-799 (2018).
11. Zhang, Y. *et al.* Emergence of Kondo lattice behavior in a van der Waals itinerant ferromagnet, Fe₃GeTe₂. *Sci. Adv.* **4**, eaao6791 (2018).
12. You, Y. *et al.* Angular dependence of the topological Hall effect in the uniaxial van der Waals ferromagnet Fe₃GeTe₂. *Phys. Rev. B* **100**, 134441 (2019).
13. Gong, C. & Zhang, X. Two-dimensional magnetic crystals and emergent heterostructure devices. *Science* **363**, eaav4450 (2019).
14. Gong, C. *et al.* Discovery of intrinsic ferromagnetism in two-dimensional van der Waals crystals. *Nature* **546**, 265-269 (2017).
15. Nagaosa, N., Sinova, J., Onoda, S., MacDonald, A. H. & Ong, N. P. Anomalous hall effect. *Rev. Mod. Phys.* **82**, 1539 (2010).

REVIEWER COMMENTS

Reviewer #1 (Remarks to the Author):

I appreciate the authors' efforts in providing more experimental progress and necessary discussions on the origin of skin characteristics. As to their replies to my two major concerns, I have the following comments.

1. I understand the difficulty in fabrications of the JJs varied in such narrow channel lengths under the premise of other conditions unchanged, and they did give a decay tendency with increasing channel length up to 450nm. The summary of the parameters in Table-R1 sounds not so satisfactory to me since the analysis of the fitted decay length and calculations is definitely more favorable to validate the mechanisms of triplet pair correlations in their systems, just as their 3rd answer to Reviewer#2. Nevertheless, that really needs some demanding experiments.
2. They did include some relevant discussions on how the skin feature comes, but indeed no direct evidence was shown to 'claim' the topological nature of Fe₃GeTe₂ by any other means except for the double-slit interference patterns in this manuscript. I could accept that they take it as a possible scenario, but the existence of the surface state in Fe₃GeTe₂ is still unclear as lacking some solid experiment proofs, and the statement should be more careful.
3. The comparison of the offsets in $I_c(B)$ with magnetic anisotropy between out-of-plane and in-plane might be evidenced by the magnified SI-Fig.7 near the zero magnetic fields, as the maximal I_c is reached when the net magnetic moment is zero.

Reviewer #2 (Remarks to the Author):

The new data, analyses and discussions have clearly improved the overall quality of this manuscript, and most of my concerns from the previous review have been properly addressed. Thus I would recommend the revised manuscript for publication in Nature Communications with a couple of final/minor suggestions as follows.

- 1) It would be great if the authors could add the characteristic voltage V_c , denoted as $I_c R_n$, versus L_j plot as the inset of Fig. 2(c), from which one can more reliably see the long-range nature of proximity-induced (spin-triplet) Cooper pairs in the Fe₃GeTe₂ Josephson barrier.
- 2) I feel that data suggesting the short-range Josephson supercurrents across a Fe_{0.25}TaS₂ barrier need to be presented in the main text.

Reviewer #3 (Remarks to the Author):

The authors have improved the manuscript considerably in response to the comments by the three reviewers. I still noticed a few minor issues that should be fixed before publication.

One of my previous comments was this:

5) The last sentence before the Discussion reads, "The $I_c(B)$ at the in-plane magnetic field has a larger offset than at the out-of-plane magnetic field, consistent with perpendicular magnetic anisotropy of the Fe₃GeTe₂ layer [45,46]." That argument sounds backwards to me. Perpendicular magnetic anisotropy should cause a larger magnetic remanence, and larger hysteresis, when the field is oriented out-of-plane.

Here is the authors' reply:

"As shown in Supplementary Fig. 7, the Fe₃GeTe₂ has perpendicular anisotropy, and its easy axis of magnetization is along the c-axis direction (out-of-plane). When the magnetic field in the positive direction scans to the negative magnetic field above the saturation magnetic field, the magnetic

moment of the Fe₃GeTe₂ will reverse at the applied negative magnetic field beyond the coercive field of the Fe₃GeTe₂. When the applied magnetic field is higher than the saturation magnetic field, the magnetic moments will align in the direction of the negative magnetic field. Since the in-plane saturation magnetic field is larger than the out-of-plane one, a larger negative magnetic field is required to flip the magnetic moment of the Fe₃GeTe₂, which can cause a larger hysteresis phenomenon. Therefore, the $I_c(B)$ in the in-plane magnetic field has a larger offset than in the out-of-plane magnetic field. We have updated the information in our revised manuscript to clarify this point." Here are the problems with that explanation: The hysteresis in $I_c(B)$ depends on the remanent magnetization, not the saturation field, so the authors' focus on saturation field is not relevant. But here is the most important point: Supplementary Figure 7(b) shows NO HYSTERESIS in the M vs B data. That is true for both orientations of the applied field – in-plane and out-of-plane. (Zero hysteresis means that the coercive field is exactly zero; so that part of the authors' response to my comment is also not relevant.) The lack of hysteresis with the field oriented out-of-plane is probably due to the formation of up- and down-pointing domains, as shown in the MFM data in Figure R2(c) in the authors response to the reviewers. I could not find Figures R2(b) and (c) anywhere in the manuscript or the Supplementary Material; they should be there. In any case, one must conclude that the magnetic properties of the very small Fe₃GeTe₂ flakes used in the experiment are different from those of the bulk sample used in the measurements shown in Supplementary Figure 7(b). That could be due to the proximity with the superconductor. The bottom line is that I am still not satisfied with this statement in the manuscript: "The $I_c(B)$ at the in-plane magnetic field has a larger offset than at the out-of-plane magnetic field, consistent with perpendicular magnetic anisotropy of the Fe₃GeTe₂ layer^{43,44}."

Here are a few additional minor points:

- i) At the bottom of page 4 and top of page 5, the second transition at 5.4 K is attributed to "the proximity-induced superconducting transition in the S/F/S." It is probably the transition of the S/F bilayers at each end of the S/F/S.
- ii) Page 6 bottom: "The supercurrent density along the z-axis (extracted from Fig. 3e) is relatively uniform, but with two pronounced peaks (Fig. 3f), indicating surface-dominated transport along the top and bottom Fe₃GeTe₂ surfaces." Shouldn't that be the "surface-dominated transport along the two edges of the Fe₃GeTe₂ strip"? The supercurrent along the top and bottom surfaces is shown in Figure 3(c).
- iii) At the end of the first paragraph of the Discussion, "exiting" should be "exciting".

Responses to reviewers' comments

(MS# NCOMMS-22-25728A)

We thank all three reviewers for investing their valuable time and effort on reviewing our manuscript and we appreciate the comments that helped us to clarify, strengthen, and improve our manuscript. We believe that our modifications have fully addressed the referees concerns and our revised manuscript is currently organized in such a way to convey more effectively our central findings. Based on the responses below, we have also updated one figure in the revised manuscript, three figures in the Supplementary Information, in response to the comments. Our point-by-point responses to the referees are as follows.

Reviewer #1

1. Comment: I understand the difficulty in fabrications of the JJs varied in such narrow channel lengths under the premise of other conditions unchanged, and they did give a decay tendency with increasing channel length up to 450 nm. The summary of the parameters in Table-R1 sounds not so satisfactory to me since the analysis of the fitted decay length and calculations is definitely more favorable to validate the mechanisms of triplet pair correlations in their systems, just as their 3rd answer to Reviewer#2. Nevertheless, that really needs some demanding experiments.

Response:

The mechanically exfoliated Fe_3GeTe_2 and NbSe_2 with irregular shapes can cause different cross-sectional areas in the different devices. As a result, it may not be accurate to draw the L_j -dependent superconducting critical current I_c plots. However, there is an alternative way to demonstrate the L_j -dependent behavior of the Josephson junctions: The characteristic voltage $I_c R_n$ product as an important parameter of Josephson junction has been reported to demonstrate the strength of the Josephson coupling without the sample-specific geometrical factors and can be utilized to estimate a decay length of Josephson supercurrents¹⁻⁴. Therefore, in our case, the characteristic voltage $I_c R_n$ product can be employed to estimate our junction decay length. We have theoretically calculated that the spin-triplet supercurrents decay length is ~ 40.7 nm which is limited

by a thermal coherence length ζ (see Supplementary Note 2). The mean-free path l in Fe_3GeTe_2 layer is significantly shorter than the coherence length ζ and we can take our Josephson junction into “dirty limit” $l < \zeta$.^{5,6} As a result, to qualitatively demonstrate the decay tendency in the Fe_3GeTe_2 spacer, we fit the L_j -dependent characteristic voltage $I_c R_n$ product using an exponential decay function, $\exp(-\frac{L_j}{\zeta})$, which is appropriate in the “dirty limit” junction regime to estimate the decay length of Josephson supercurrents through the Fe_3GeTe_2 barrier approximately⁶⁻⁹. As shown in Fig. R1, the estimated $\zeta = 227 \pm 18$ nm at 3 K under zero magnetic field is much longer than the theoretically calculated coherence length, but comparable to our experimental results. However, this result can only be a rough estimate, and more accurate quantitative research indeed demands more rigorous experimental techniques in future studies, as the referee commented. We have added the descriptions and the figure in our revised manuscript.

2. Comment: They did include some relevant discussions on how the skin feature comes, but indeed no direct evidence was shown to ‘claim’ the topological nature of Fe_3GeTe_2 by any other means except for the double-slit interference patterns in this manuscript. I could accept that they take it as a possible scenario, but the existence of the surface state in Fe_3GeTe_2 is still unclear as lacking some solid experiment proofs, and the statement should be more careful.

Response: We thank the reviewer for this helpful caution. We agree with the reviewer that the explanation based on the topological surface states of Fe_3GeTe_2 is a pure theoretical speculation, and the present experimental results are unable to provide a smoking-gun evidence for this mechanism. In response to this situation and in accordance with the suggestions of the reviewer, we have revised the discussion section concerning the possible mechanisms behind the skin features.

Firstly, we revise the description of the possible mechanisms with a more reserved statement of their relevance to the current experimental systems. We provide a more precise understanding of these mechanisms by emphasizing their theoretical rooting.

Secondly, in the revised discussions we explicitly state that precise nature of the surface state in Fe_3GeTe_2 is still an open issue lacking some solid experiment proofs in the present work, and we discuss the experimental directions of pinning down these theoretical mechanisms. For details of these revision, please see the revised manuscript and summary of changes.

Finally, we wish to note that while the surface state in Fe_3GeTe_2 is indeed still unclear and lacks solid experiment proofs, the physical principle behind the proposal of this surface state is reasonable because it mostly relies on the ferromagnetism and the mirror symmetry breaking on the surface of the Fe_3GeTe_2 , which is evidently the case for the current experimental system. However, we fully agree with the reviewer that these surface states require more solid experiment proofs, and this is certainly an exciting direction for further studies.

3. Comment: The comparison of the offsets in $I_c(B)$ with magnetic anisotropy between *out-of-plane* and *in-plane* might be evidenced by the magnified SI-Fig.7 near the zero magnetic fields, as the maximal I_c is reached when the net magnetic moment is zero.

Response: The magnified SI-Fig.7 near the zero magnetic fields has been shown in the inset of Figure R2. The external magnetic fields corresponding to the location where the net magnetic moment of Fe_3GeTe_2 reaches zero under the *in-plane* and the *out-of-plane* magnetic fields exhibit almost the same value of ~ 250 Oe. However, Figure R2 shows the magnetism of the bulk Fe_3GeTe_2 instead of the Fe_3GeTe_2 flakes that are utilized in our lateral junctions. Actually, the magnetism of Fe_3GeTe_2 flake is thickness dependent. When the thickness of the Fe_3GeTe_2 flake decreases, the flake coercive field increases¹⁰. For a layer of the Fe_3GeTe_2 flake, the anomalous Hall resistance changes sharply near the coercive field, and the hysteresis loop shows a nearly rectangular shape under the *out-of-plane* magnetic field. The thinner the Fe_3GeTe_2 is, the closer the ratio of remnant magnetization to saturation magnetization (M_R/M_S) is to 1, which shows the hard magnetic property¹⁰. Regarding our Fe_3GeTe_2 flakes, we studied the anomalous Hall resistance of the Fe_3GeTe_2 flake with a thickness of 10 nm under different magnetic field orientations as shown in Figure R3a. Figure R3b is a Hall effect curve

with the *in-plane* external magnetic field perpendicular to the current. It shows that the coercive field of the Fe₃GeTe₂ flake under the *out-of-plane* magnetic field is 0.38 T, which is far less than that of 1.18 T under the *in-plane* magnetic field. However, to more rigorously present our $I_c(B)$ offset data, we removed the comparison with the *out-of-plane* and *in-plane* coercive field of the Fe₃GeTe₂ flake in our main manuscript. In addition, as we mentioned earlier, the superconductivity can weaken the magnetization of the Fe₃GeTe₂ flake through the proximity effect¹¹. Therefore, the value of offset in $I_c(B)$ could be much smaller than the coercive field of the pristine Fe₃GeTe₂ flake. We have added the Figure R3 in our revised Supporting Information.

Reviewer #2

1. Comment: It would be great if the authors could add the characteristic voltage V_c , denoted as $I_c R_n$, versus L_j plot as the inset of Fig. 2(c), from which one can more reliably see the long-range nature of proximity-induced (spin-triplet) Cooper pairs in the Fe₃GeTe₂ Josephson barrier.

Response:

By using the geometry-independent characteristic voltage $I_c R_n$ product, we conducted the fitting of the L_j -dependent characteristic voltage $I_c R_n$ product using an exponential decay function, $\exp(-\frac{L_j}{\xi})$ and estimated the decay length of Josephson supercurrents through the Fe₃GeTe₂ barrier roughly^{1-4,6-9}. As shown in Fig. R1, the estimated $\xi = 227 \pm 18$ nm at 3 K under zero magnetic field is much longer than the theoretical calculated coherence length, but comparable to our experimental results. However, this result can only be a rough estimate, and more accurate quantitative research requires more rigorous experimental techniques in the future studies. We have added the descriptions and the figure as the inset of Fig. 2(c) in our revised manuscript.

2. Comment: I feel that data suggesting the short-range Josephson supercurrents across a Fe_{0.25}TaS₂ barrier need to be presented in the main text.

Response: We have added the related descriptions about the Fe_{0.25}TaS₂ in the main text. To prove that the long-range Josephson supercurrents is directly related to the nature of

Fe₃GeTe₂, we replace the Fe₃GeTe₂ layer with a ferromagnet of Fe_{0.25}TaS₂ as a barrier to construct NbSe₂/Fe_{0.25}TaS₂/NbSe₂ lateral junction with the channel length of 200 nm (Supplementary Fig. 2). There is no Josephson supercurrent observed in this NbSe₂/Fe_{0.25}TaS₂/NbSe₂ junction with the 200 nm channel length.

Reviewer #3

1. Comment: The authors have improved the manuscript considerably in response to the comments by the three reviewers. I still noticed a few minor issues that should be fixed before publication.

One of my previous comments was this:

5) The last sentence before the Discussion reads, “The $I_c(B)$ at the *in-plane* magnetic field has a larger offset than at the *out-of-plane* magnetic field, consistent with perpendicular magnetic anisotropy of the Fe₃GeTe₂ layer [45,46].” That argument sounds backwards to me. Perpendicular magnetic anisotropy should cause a larger magnetic remanence, and larger hysteresis, when the field is oriented *out-of-plane*.

Here is the authors’ reply:

“As shown in Supplementary Fig. 7, the Fe₃GeTe₂ has perpendicular anisotropy, and its easy axis of magnetization is along the c-axis direction (*out-of-plane*). When the magnetic field in the positive direction scans to the negative magnetic field above the saturation magnetic field, the magnetic moment of the Fe₃GeTe₂ will reverse at the applied negative magnetic field beyond the coercive field of the Fe₃GeTe₂. When the applied magnetic field is higher than the saturation magnetic field, the magnetic moments will align in the direction of the negative magnetic field. Since the *in-plane* saturation magnetic field is larger than the *out-of-plane* one, a larger negative magnetic field is required to flip the magnetic moment of the Fe₃GeTe₂, which can cause a larger hysteresis phenomenon. Therefore, the $I_c(B)$ in the *in-plane* magnetic field has a larger offset than in the *out-of-plane* magnetic field. We have updated the information in our revised manuscript to clarify this point.”

Here are the problems with that explanation: The hysteresis in $I_c(B)$ depends on the remanent magnetization, not the saturation field, so the authors’ focus on saturation

field is not relevant. But here is the most important point: Supplementary Figure 7(b) shows NO HYSTERESIS in the M vs B data. That is true for both orientations of the applied field – *in-plane* and *out-of-plane*. (Zero hysteresis means that the coercive field is exactly zero; so that part of the authors' response to my comment is also not relevant.) The lack of hysteresis with the field oriented *out-of-plane* is probably due to the formation of up- and down-pointing domains, as shown in the MFM data in Figure R2(c) in the authors response to the reviewers. I could not find Figures R2(b) and (c) anywhere in the manuscript or the Supplementary Material; they should be there.

In any case, one must conclude that the magnetic properties of the very small Fe_3GeTe_2 flakes used in the experiment are different from those of the bulk sample used in the measurements shown in Supplementary Figure 7(b). That could be due to the proximity with the superconductor.

The bottom line is that I am still not satisfied with this statement in the manuscript: “The $I_c(B)$ at the *in-plane* magnetic field has a larger offset than at the *out-of-plane* magnetic field, consistent with perpendicular magnetic anisotropy of the Fe_3GeTe_2 layer^{43,44}.”

Response: The hysteresis in $I_c(B)$ depends on the remanent magnetization of Fe_3GeTe_2 flake, and the maximal I_c is reached when the net magnetic moment is zero. The magnified SI-Fig.8 near the zero magnetic fields has been shown in the inset of Figure R2. The external magnetic fields corresponding to the location where the net magnetic moment of Fe_3GeTe_2 reaches zero under the *in-plane* and *out-of-plane* magnetic fields exhibit almost the same value of ~ 250 Oe. However, the Figure R2 shows the magnetism behavior of the bulk Fe_3GeTe_2 instead of Fe_3GeTe_2 flakes that are utilized in our lateral junction devices. Actually, the magnetism of Fe_3GeTe_2 flake is thickness dependent. When the thickness of the Fe_3GeTe_2 sample decreases, the coercive field increases¹⁰. For a layer of the Fe_3GeTe_2 flake, the anomalous Hall resistance changes sharply near the coercive field, and the hysteresis loop shows a nearly rectangular shape under the *out-of-plane* magnetic field. The thinner the Fe_3GeTe_2 is, the closer the ratio of remanent magnetization to saturation magnetization (M_R/M_S) is to 1, which shows the hard magnetic property¹⁰. Regarding our Fe_3GeTe_2 flakes, we studied the

anomalous Hall resistance of the Fe₃GeTe₂ flake with a thickness of 10 nm under different magnetic field orientations as shown in Figure R3a. Figure R3b is a Hall effect curve with the *in-plane* external magnetic field perpendicular to the current. It shows that the coercive field of the Fe₃GeTe₂ flake under the *out-of-plane* magnetic field is 0.38 T, which is far less than that of 1.18 T under the *in-plane* magnetic field. However, to more rigorously present our $I_c(B)$ offset data, we removed the comparison with the *out-of-plane* and *in-plane* coercive field of the Fe₃GeTe₂ flake in our main manuscript. In addition, as we mentioned earlier, the superconductivity can weaken the magnetization through the proximity effect¹¹. The magnetic force microscopy (MFM) data in Figure R4 also indicates that the superconductivity can weaken the magnetization of the Fe₃GeTe₂ flake. Therefore, the value of offset in $I_c(B)$ of our lateral junction could be much smaller than the coercive field of the pristine Fe₃GeTe₂ flake. We have added the Figure R3 in our revised Supporting Information.

2. Comment: Here are a few additional minor points: i) At the bottom of page 4 and top of page 5, the second transition at 5.4 K is attributed to “the proximity-induced superconducting transition in the S/F/S.” It is probably the transition of the S/F bilayers at each end of the S/F/S.

Response: We have revised the sentences “the proximity-induced superconducting transition in the S/F/S (5.4 K)” into “the proximity-induced superconducting transition of the S/F bilayers at each end of the S/F/S (5.4 K)”. We have also carefully checked other English writing in the revised manuscript.

3. Comment: ii) Page 6 bottom: “The supercurrent density along the z-axis (extracted from Fig. 3e) is relatively uniform, but with two pronounced peaks (Fig. 3f), indicating surface-dominated transport along the top and bottom Fe₃GeTe₂ surfaces.” Shouldn’t that be the “surface-dominated transport along the two edges of the Fe₃GeTe₂ strip”? The supercurrent along the top and bottom surfaces is shown in Figure 3(c).

Response: We have revised the sentences “indicating surface-dominated transport along the top and bottom Fe₃GeTe₂ surfaces” into “indicating surface-dominated

transport along the two edges of the Fe_3GeTe_2 strip” in our revised manuscript.

4. Comment: iii) At the end of the first paragraph of the Discussion, “exiting” should be “exciting”.

Response: We have revised the word “exiting” into “exciting” in our revised manuscript.

Figure R1 Characteristic voltage $I_c R_n$ as a function of L_j , and the decay length of the Josephson coupling through the Fe_3GeTe_2 barrier is fitted by an exponential decay function (red curve).

Figure R2 Magnetic properties of bulk Fe_3GeTe_2 . (a) Temperature-dependent magnetization curves measured at 1000 Oe external field along both c-axis and ab-plane for as-grown Fe_3GeTe_2 with a Curie temperature of 200 K. (b) Field-dependent magnetization plots of Fe_3GeTe_2 at 3 K. The external field is applied in different directions of $B//c$ (red) and $B//ab$ (blue), and the inset is the enlarged part in the dashed box.

Figure R3 (a) The anomalous Hall resistance of the Fe_3GeTe_2 flake with thickness of 10 nm under *out-of-plane* magnetic field, and the coercive field B_c is 0.39 T. (b) The anomalous Hall resistance of the Fe_3GeTe_2 flake with thickness of 10 nm under *in-plane* magnetic field, and the coercive field B_c is 1.18 T.

Figure R4 (a) Schematic illustrate of FGT/NbSe₂ heterostructure characterized by MFM. (b) Magnetization distribution of the FGT layer across a boundary region of the heterostructure with and without the underlying NbSe₂ by MFM at 0 T magnetic field.

References

1. Kontos, T. *et al.* Josephson junction through a thin ferromagnetic layer: negative coupling. *Phys. Rev. Lett.* **89**, 137007 (2002).
2. Kim, M. *et al.* Strong proximity Josephson coupling in vertically stacked NbSe₂ – graphene – NbSe₂ van der Waals junctions. *Nano Lett.* **17**, 6125-6130 (2017).
3. Kang, K. *et al.* Van der Waals π Josephson junctions. <https://doi.org/10.48550/arXiv.2201.09185>.
4. Tinkham, M. *Introduction to Superconductivity* (Dover, 2004).
5. Bozovic, I. *et al.* Giant proximity effect in cuprate superconductors. *Phys. Rev. Lett.* **93**, 157002 (2004).
6. Barone, A. & Paterno, G. *Physics and Applications of the Josephson Effect* 2nd edn (John Wiley & Sons, 1982).
7. Bell, C. *et al.* Proximity and Josephson effects in superconductor/antiferromagnetic Nb/ γ -Fe₅₀Mn₅₀ heterostructures. *Phys. Rev. B* **68**, 144517 (2003).
8. Weides, M., Disch, M., Kohlstedt, H. & Bürgler, D. Observation of Josephson coupling through an interlayer of antiferromagnetically ordered chromium. *Phys. Rev. B* **80**, 064508 (2009).
9. Jeon, K.-R. *et al.* Long-range supercurrents through a chiral non-collinear antiferromagnet in lateral Josephson junctions. *Nat. Mater.* **20**, 1358-1363 (2021).
10. Tan, C. *et al.* Hard magnetic properties in nanoflake van der Waals Fe₃GeTe₂. *Nat. Commun.* **9**, 1554 (2018).
11. Buzdin, A. I. Proximity effects in superconductor-ferromagnet heterostructures. *Rev.*

Mod. Phys. **77**, 935 (2005).

REVIEWERS' COMMENTS

Reviewer #1 (Remarks to the Author):

The authors have answered most of my questions. It is now ready for the final publication.

Reviewer #3 (Remarks to the Author):

The authors have improved the manuscript again in the second revision. The Discussion section, in particular, has been improved considerably. I recommend that it be published in Nature Communications.

I found two minor errors that the authors may wish to correct before publication:

- 1) On page 3, line 40, the authors cite references [18-20] as evidence for spin-triplet pairing. References [18] and [19] are appropriate here, but [20] is not. It is an important paper, but it does not show any evidence for spin-triplet pairing. Quite the contrary – the supercurrent through the FeMn barrier studied in that work has an extremely short range. If the authors don't want to renumber all of their references, there are many other spin-triplet works in the literature they could cite instead.
- 2) At the end of Supplementary Note 2, on lines 97-99 of the SI, the authors write, "As shown in the inset of Fig. 3c, the estimated $\xi = 227 \pm 18$ nm at 3K under zero magnetic field is much longer than the theoretically calculated coherence length, but comparable to our experimental results." I believe they meant the inset of Fig. 2c rather than 3c. Also, I don't understand the last phrase in that sentence. The estimate of 227 nm is the experimental result, so why do they say it is "comparable to our experimental results"?

Responses to reviewers' comments

(MS# NCOMMS-22-25728B)

Reviewer #3

1. Comment: On page 3, line 40, the authors cite references [18-20] as evidence for spin-triplet pairing. References [18] and [19] are appropriate here, but [20] is not. It is an important paper, but it does not show any evidence for spin-triplet pairing. Quite the contrary - the supercurrent through the FeMn barrier studied in that work has an extremely short range. If the authors don't want to renumber all of their references, there are many other spin-triplet works in the literature they could cite instead.

Response: We carefully checked the cited references in the manuscript and updated the information in our revised manuscript as the reviewer suggested.

2. Comment: At the end of Supplementary Note 2, on lines 97-99 of the SI, the authors write, "As shown in the inset of Fig. 3c, the estimated $\xi = 227 \pm 18$ nm at 3K under zero magnetic field is much longer than the theoretically calculated coherence length, but comparable to our experimental results." I believe they meant the inset of Fig. 2c rather than 3c. Also, I don't understand the last phrase in that sentence. The estimate of 227 nm is the experimental result, so why do they say it is "comparable to our experimental results"?

Response: We have revised the "As shown in the inset of Fig. 3c, the estimated $\xi = 227 \pm 18$ nm at 3 K under zero magnetic field is much longer than the theoretically calculated coherence length, but comparable to our experimental results." into "As shown in the inset of Fig. 2c, the estimated $\xi = 227 \pm 18$ nm at 3 K under zero magnetic field is much longer than the theoretically calculated coherence length, but comparable to the channel length of our device."